# Unextractable Protocol Models: Collaborative Training and Inference without Weight Materialization

**Alexander Long**\*    **Chamin Hewa Koneputugodage**\*    **Thalaiyasingam Ajanthan**    **Yan Zuo**

**Gil Avraham**    **Violetta Shevchenko**    **Hadi Mohaghegh Dolatabadi**    **Sameera Ramasinghe**

Pluralis Research

## Abstract

We consider a decentralized setup in which the participants collaboratively train and serve a large neural network, and where each participant only processes a subset of the model. In this setup, we explore the possibility of *unmaterializable* weights, where a full weight set is *never* available to any one participant. We introduce Unextractable Protocol Models (UPMs): a training and inference framework that leverages the *sharded model setup* to ensure model shards (*i.e.*, subsets) held by participants are incompatible at different time steps. UPMs periodically inject time-varying, random, invertible transforms at participant boundaries; preserving the overall network function yet rendering cross-time assemblies incoherent. On Qwen-2.5-0.5B and Llama-3.2-1B, 10 000 transforms leave FP32 perplexity unchanged ($\Delta$PPL$< 0.01$; Jensen–Shannon drift $< 4 \times 10^{-5}$), and we show how to control growth for lower precision datatypes. Applying a transform every 30s adds 3% latency, 0.1% bandwidth, and 10% GPU-memory overhead at inference, while training overhead falls to 1.6% time and $< 1\%$ memory. We consider several attacks, showing that the requirements of direct attacks are impractical and easy to defend against, and that gradient-based fine-tuning of stitched partitions consumes $\geq 60\%$ of the tokens required to train from scratch. By enabling models to be collaboratively trained yet not extracted, UPMs make it practical to embed programmatic incentive mechanisms in community-driven decentralized training.

## 1 Introduction

Foundation models are trained on massively distributed infrastructures [13, 35], and there is a growing interest in scaling beyond centralized computing clusters via communication-efficient and fault-tolerant decentralized systems [11, 28, 51, 50, 47]. Even though decentralized systems may facilitate volunteer-based, multi-party training; without a way for contributors to recoup the hundreds of millions in training costs [8, 23, 41], such collaborations remain economically infeasible at large scales [1, 57]. A viable solution is to design incentivization strategies to allow value flow from model usage to the training contributors, however, this is not possible if the model is freely accessible to participants (as with a full weight set participants can independently capture model value).

To resolve this seeming contradiction, how can a model be collaboratively trained and hosted, while the weight set remains private, we introduce Unextractable Protocol Models (UPMs). UPMs utilize structural properties of the Model Parallel (MP) [17, 54] training setup, where the neural network is sharded into small subsets and split over devices (in our case, participants), and no one device accesses the full model at any time. This is a novel setting, where the model architecture, hyperparameters, and datasets can be open-sourced, yet a full weight set *never* exists in any location at any one point.

---

\*Equal contribution.

39th Conference on Neural Information Processing Systems (NeurIPS 2025).

In the decentralized case, a motivated attacker may attempt to periodically rejoin training in different stages to obtain a full copy of the model. UPMs address this by making the weights of the model shards time-dependent and incompatible across time-steps. Periodically, time-varying random invertible transforms are morphed into the model weights at the stage boundaries. These transforms cancel out at the same time step, preserving the end-to-end network function, however, they are incompatible across time steps, ensuring that stitching shards from different times results in an incoherent model.

We show that our *morphing* approach is compatible with various neural network architectures, including transformer layers [60], and discuss the efficacy of different types of transformations, showing virtually no performance degradation and negligible communication and memory overhead for inference. Furthermore, we analyze strong learning based attacks to stitch shards from different time steps, and show that recovering the full model functionality requires training costs in the same order as training the full model from scratch. Finally, we discuss how our framework affects optimization, and show that training is unaffected for a compatible optimizer.

We make the following contributions:

- We introduce *Unextractable Protocol Models* (UPMs), a framework enabling collaborative training and inference without allowing any participant to extract the full model weights.
- We provide a thorough analysis of UPMs, identifying which architectures and layer types they support, the key requirements for the transforms, how they affect optimization, and potential attack scenarios along with practical defenses.
- We empirically demonstrate on billion-parameter decoder-based language models that UPMs do not alter the model performance even with up to 10k morphing steps, and the communication and memory overhead is negligible compared to inference/training cost.

## 2 Setting: Decentralized Models and the Weight Materialization Threat

We first introduce the decentralized model setup (*i.e.*, *the protocol*), where participants collaboratively train or serve a model. To ensure economic viability, we argue that the full model weights cannot be allowed to *materialize*, motivating a design where each participant processes only a model subset. We then examine potential extraction attempts, where adversaries try to reconstruct the full weight set across time steps. The next section introduces UPMs, which prevent this by ensuring that weights across time steps form an incoherent model, while preserving exact functionality within each step.

### 2.1 Decentralized Protocols for Large Models

We consider a decentralized protocol that enables collaborative training and inference of large models by pooling participants' compute resources. This level of compute is not within the reach of a small group of individuals and therefore requires a large group of participants. The setting is inherently trustless, we can assume an honest majority but must assume there are malicious actors.

**Economic feasibility.** For such a protocol to be economically feasible, it must be able to derive value from the trained model. The key to this, explained in Section A, is to ensure that the trained model's utility is *excludable*: accessible only through the protocol. As long as excludability holds, access can be priced and revenue shared among participants.

To achieve excludability, we propose guaranteeing *unextractability*: preventing participants from being able to extract the full weight set. This means the full model *only exists within the protocol*, preventing weight materialization for any participant. We show that unextractability can be ensured (with high probability) with a lightweight construction that uses the structure of the model setup, as opposed to using expensive cryptographic primitives. Notably, code and data can remain open, enabling open-source development of large models while preserving economic feasibility.

We also need to consider *partial excludability*, where an attacker can produce a model $M'$ with equivalent performance to the protocol model $\hat{M}$ using information gained during participation (see Section A). We quantify excludability by defining the *excludability ratio* $\mathcal{E}_r = \frac{\mathcal{C}(M')}{\mathcal{C}(\widehat{M})}$, where $\mathcal{C}(\widehat{M})$ is the (extremely large) compute required to train $\hat{M}$ in the protocol, and $\mathcal{C}(M')$ is compute required to produce $M'$ (both participating in the protocol and extra training). If $\mathcal{E}_r$ is very low, so $\mathcal{C}(M')$ is within the reach of a small group of individuals, the protocol is not economically viable.

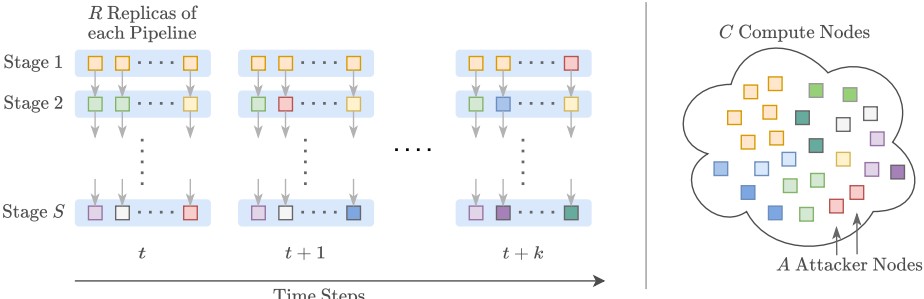

Figure 1: A pipeline parallel setup with $R$ pipelines and $S$ stages, where each stage is occupied by one of $C$ compute nodes. Participants can operate multiple nodes within a stage (color indicates participant identity). Nodes in red indicate an attacker, who hides their identity to try and get access to different stages of the model over time.

**Model setup.**  We require each participant to only hold and processes a subset of the model. This can be done through Model Parallelism (MP) [14, 54], which divides the model into shards distributed across devices. We consider a special case of MP, Pipeline Parallelism (PP) [14, 17], where each shard contains a consecutive block of layers called a pipeline stage. Participants only communicate with adjacent stages, avoiding the costly all-to-all communication common in other MP strategies.

Assume the model $F : \mathcal{X} \to \mathcal{Y}$ is divided into $S$ sequential pipeline stages. The model computes

$$F(X_0) = f_S \circ f_{S-1} \circ ... \circ f_1(X_0) \,, \qquad X_i = f_i(X_{i-1}; \theta_i) \,, \quad \text{for } i = 1, \ldots, S \,, \qquad (1)$$

where $X_i$ are the intermediate model outputs (*i.e.*, activations) with $X_0$ as the input to the model, and stage $f_i$ has parameters $\theta_i$. The protocol maintains $R$ replicas of each pipeline, both allowing to scale for more participants and compute (data parallelism) and ensuring fault tolerance (stages are not lost if a participant drops out).

Suppose a compute pool of $C > RS$ nodes from all participants. The protocol randomly allocates nodes to each of the $RS$ slots, each a specific replica of a stage, without revealing the stage identity. The protocol can implement stricter measures, such as ensuring that a participant can only hold slots from a single stage. We assume that the total required compute for the protocol is much larger than any participant can provide, so any attacker (or attack coalition) has total compute $A$ where $A \ll RS < C$. Thus, an attacker cannot have access to all stages at any given time. However, they can hide their identity and access different stages over multiple time steps, which we discuss below. Our overall setup is illustrated in Figure 1.

## 2.2   Threat of Weight Materialization

Let us consider how an attacker (or attack coalition) can attempt to extract the full model weights over $T$ time steps. Note that the attacker controls a small fraction $p = \frac{A}{C} \ll 1$ of all nodes, so the probability of the protocol allocating a particular slot to the attacker is approximately $p$.

For the attacker to be successful, the attacker must access at least one slot from each of the $S$ stages. If an attacker has multiple nodes, they can masquerade as multiple participants simultaneously to try and access multiple stages. Since $p \ll 1$, it is highly unlikely that the attacker will be assigned to every stage, in fact it is likely that $A < S$. However, by repeatedly leaving the protocol and rejoining as a new participant, the attacker may gain access to all stages over many time steps. The probability of this happening within $T$ time steps is approximately $(pRT)^S$ (derived in Section C). Note that training and inference require an extremely large number of time steps, so even a small $p$ can ensure a successful attack once $T \approx \frac{1}{pR}$.

Such *piecewise Sybil attacks* pose a threat of weight materialization. One can try to deter this via centralized authentication systems or cryptographic solutions [12, 53], but they require additional constraints on the protocol and incur prohibitive computation and/or communication overhead at scale. Instead, we leverage the sharded model structure itself to enforce weight unmaterializability.

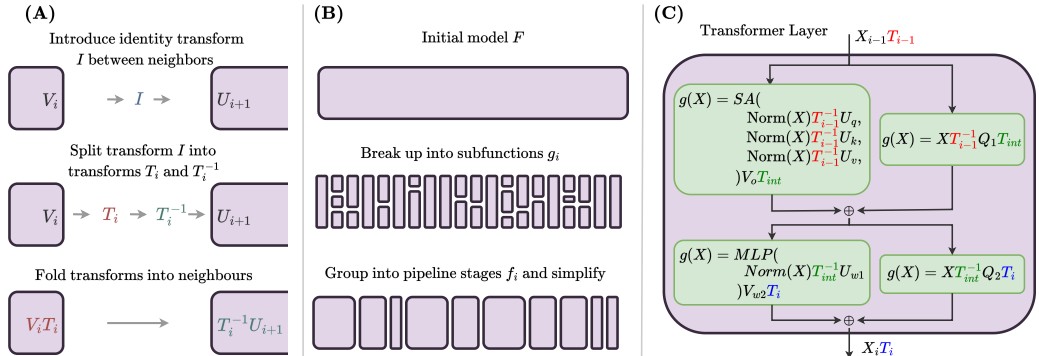

Figure 2: Our framework ensures stage weights vary over time by folding in transforms, yet the end-to-end function is preserved. **(A)** Our transform inducing process. **(B)** Our framework reasons about subfunctions $g_i$, which are grouped into pipeline stages. **(C)** A transformer layer as a stage with transforms applied. Note that every weight is morphed: $T_{int}$ must align with $T_{i-1}$ and $T_i$ due to the skip connection, meaning compatibility at a stage boundary also depends on the other boundary.

## 3 Unextractable Protocol Models

To prevent an attacker from stitching together weights collected at different times, we make the weights time-dependent while ensuring the end-to-end network function is unchanged at any point in time. We achieve this by periodically morphing the weights with transformations that cancel out, thus weights collected at different times are incoherent as their integrated transforms do not cancel out. Note that the transforms are repeatedly applied to weights without removing old transforms, and knowledge of the transform is discarded after the morphing.

Every *morphing step*, where transforms are folded into the model weights, defines a new *transform time step*[2] in the protocol. Consider two neighboring stages $f_i$ and $f_{i+1}$ with weights $\theta_i(t)$ and $\theta_{i+1}(t)$ at transform time step $t$. We morph weights so that $\theta_i(t)$ and $\theta_{i+1}(t)$ remain compatible, but $\theta_i(t)$ and $\theta_{i+1}(t')$ for $t \neq t'$ are incompatible. As shown in Figure 2 (A), this is done by introducing an identity function between the neighbors, decomposing it into a random transform and its inverse, and then folding them into the weights of the neighbors. This operation leaves the end-to-end function $F$ unchanged, though at each time step $t$ the intermediate activations between stages lie in a different space. Consequently, $\theta_i(t)$ and $\theta_{i+1}(t')$ are incompatible, as $\theta_{i+1}(t')$ is only compatible with activations produced by $\theta_i(t')$. We now formalize this process.

### 3.1 Approach

The key to our approach is to identify which weight matrices need transforms applied to them and how to apply those transforms. This involves decomposing the model into smaller *subfunctions* that allow transforms to be applied between them.

**Valid subfunctions.** We first define a *valid* subfunction as a subfunction of the form

$$g_i(X) = \Phi_i(XU_i)V_i \tag{2}$$

where $U_i$ and $V_i$ are weight matrices, which we call the *entry* and *exit* matrices of the subfunction, and $\Phi_i$ is any (possibly nonlinear) function. Let $d$ denote the output dimension of $g_i$.

We now show how to apply transforms between valid subfunctions so they become incompatible across time steps. Consider two consecutive valid subfunctions $g_i = \Phi_i(XU_i)V_i$ and $g_{i+1} = \Phi_{i+1}(XU_{i+1})V_{i+1}$, composed as $g_{i+1} \circ g_i$. As shown in (see Figure 2 (A)), this is done by

1. inserting an identity transforma between them: $g_{i+1} \circ g_i = g_{i+1} \circ I \circ g_i, \ I(X) = XI_d$,
2. splitting it into a random transform and its inverse: $I(X) = XTT^{-1}, T \in \mathrm{GL}(d, \mathbb{R})$[3], and
3. folding the transforms into the weights: $V_i \leftarrow V_iT, \ U_{i+1} \leftarrow T^{-1}U_{i+1}$.

---

[2]This time step may be different from the time step of inference calls or training iterations.

[3]Group of $d \times d$ invertible matrices over $\mathbb{R}$

This procedure is called a morphing step and advances the protocol a transform time step. We index components by time step, e.g., $V_i(t) = V_i(t-1)T_i(t)$. In practice, the morphing step between $t-1$ and $t$ consists of: $g_i$ generates the random transform pair $T(t)$ and $T(t)^{-1}$, sends it to $g_{i+1}$, both fold them into their weights and then discard them. Since the transforms are discarded, we consider them to be *ephemeral*, they are baked into the weights and there is no other knowledge of them.

To see why this causes cross-time incompatibility, assume we have access to $g_i$ at time $t$ and $g_{i+1}$ at time $t'$ with $t < t'$. Then we have access to $V_i(t)$, $T_i(t)$, $U_{i+1}(t')$ and $T_i(t')$. However, $V_i(t)$ and $U_{i+1}(t')$ cannot be used together: if $g_i(X) = Z$ before any transforms are applied, then using $V_i(t)$ within $g_i$ yields $ZT_i(1)...T_i(t)$ but $U_{i+1}(t')$ expects $ZT_i(1)...T_i(t')$. Thus, $V_i(t)$ and $U_{i+1}(t')$ are inconsistent, their composition no longer preserves $g_{i+1} \circ g_i$. Aligning the weights requires the *bridge matrix* $\hat{T} = T_i(t+1)...T_i(t')$, but we only know the last term $T_i(t')$, so we only know the bridge matrix for $t' = t + 1$. For $t' > t + 1$, aligning the weights requires guessing a random invertible matrix. Finally, to prevent attackers from rejoining the protocol within consecutive time steps, the protocol enforces a two-step joining delay.

**Allowing identity entry/exit matrices.** As $\Phi_i$ can be anything, the main constraint in Equation (2) is the need for there to exist entry and exit weight matrices. We now relax this constraint so that at a boundary $g_i$, $g_{i+1}$, only one of $V_i$, $U_{i+1}$ need to be a weight matrix; the other we can form as an explicit identity matrix that gets morphed. For example, if our subfunction does not have a weight matrix $V_i$, then we replace it with $Q_i$ where $Q_0 = I_d$. $Q_i(t)$ and $U_{i+1}(t')$ will remain incompatible and require a bridge matrix. However, we cannot have both $V_i$ and $U_{i+1}$ as identity matrices: although $Q_i(t)$ and $Q_{i+1}(t')$ stay incompatible, they contain no information, allowing an attacker to remove them without altering the function.

Many neural network components can be expressed as compositions of these subfunctions. For example, MLPs can be written as compositions $g_i(X) = \sigma(XU_i) = \sigma(XU_i)Q_i$ with element-wise activation function $\sigma$, or equivalently $g_i(X) = \sigma(X)V_i = \sigma(Q_iX)V_i$.

**Extending to complex components.** We extend valid subfunctions to multiple inputs and outputs by generating a transform at each exit matrix, reusing transforms if the exit matrices map to the same input (or have the same dimension). Examples include RNN blocks $H_{t+1} = \Phi_h(H_tU_{h1} + X_tU_{h2})V_{h1}$, $X_{t+1} = \Phi_o(H_tU_{o1} + X_tU_{o2})V_{o2}$, or self attention $\text{SA}(X) = \Phi(XU_Q, XU_K, XU_V)V$ (here $\Phi$ is multi-headed attention). Subfunctions can also be combined: scaling in depth by composing multiple subfunctions, scaling in breadth by applying in parallel like skip connections: $(\Phi(XU_i)V_i, XQ_i)$, or multiplying them for gating: $\sum_k G_k(XU_{k,1})\Phi_k(XU_{k,2})V_k$.

**Generalizing subfunctions.** Ultimately, what matters is embedding the inverse transform into the next subfunction's weights while preserving the overall composition, leading to subfunctions of the form $g_i(X) = \Phi_i(\Omega(X)U_i)V_i$ where $g_i(XT^{-1}) = \Phi_i(\Omega(X)U_i')V_i$ for some morphing function $U_i \mapsto U_i'$ that involves $T^{-1}$ and preserves the shape of $U_i$. For example, convolution layers can be expressed as $g_i(X) = \Phi_i(\text{im2col}(X)U_i)$ where $X \in \mathbb{R}^{\cdots \times d_{i-1}}$, $\text{im2col}(X) \in \mathbb{R}^{\cdots \times Pd_{i-1}}$, $U_i \in \mathbb{R}^{(Pd_{i-1}) \times d}$ and $P$ is the convolution patch size. Thus for $T^{-1} \in \mathbb{R}^{d_{i-1} \times d_{i-1}}$, $\text{im2col}(XT^{-1})U_i = \text{im2col}(X)(I_P \otimes T^{-1})U_i$, so the inverse should be folded into $U_i$ as $(I_P \otimes T^{-1})U_i$.

**Normalization Layers.** Normalization layers pose a challenge, as they alter subfunctions to be of the form $\Phi(\text{Norm}(X)U_i(t))V_i$. The issue is that this is a non-linear transformation of $X$ before the matrix multiplication with $U_i$, so our previous strategy does not work. However, we propose the following workaround for RMSNorm layers, which have been shown to match or exceed LayerNorm and BatchNorm performance across many tasks while also being more efficient [67]. Given $X \in \mathbb{R}^{b \times d_{i-1}}$ with rows/instances $x^{(j)}$, RMSNorm is of the form

$$\text{RMSNorm}(X) = \text{norm}(X)X\text{diag}(w), \qquad \text{norm}(X) = \text{diag}_{j=1}^b\left(\left(d^{-1}\|x^{(j)}\|^2\right)^{-1/2}\right) \quad (3)$$

where $\text{norm}(\cdot)$ applies row-wise RMS normalization and $w \in \mathbb{R}^d$ are the feature scaling parameters. We introduce a transform accumulation matrix into the normalization: $\text{norm}(X) \to \text{norm}(XQ_i(t))$, with $Q_i(0)$ initialized as an orthogonal matrix. Then transforms are also folded into $Q_i(t)$, hiding them and $Q(0)$, while preserving the norm due to the orthogonality of $Q(0)$. We also absorb $\text{diag}(w)$ into $U_i$ so that transforms can be folded into the latter, $U_i(0) = \text{diag}(w)U_i$ (during training this is equivalent to removing $w$). We now have the desired functional equivalence (see Section D for

details), after morphing step $Q_i \to T^{-1}Q_i$ and $U_i \to T^{-1}Q_i$

$$f_i\left(X_{i-1}T\right) = \Phi\left(\text{norm}\left(X_{i-1}TQ_i\right)X_{i-1}TU_i\right)V_i \qquad (4)$$

$$= \Phi\left(\text{RMSNorm}(X_{i-1})X_{i-1}U_i\right)V_i. \qquad (5)$$

**Full model morphing.** Having established how to apply transforms between neural network components, we now consider the overall model. As shown in Figure 2 (B), we decompose our model into such subfunctions, and group contiguous subfunction blocks into pipeline stages. Within each stage, redundant transforms are removed.

**Extraction probabilities.** Our construction makes certain stage weights—specifically the entry/exit matrices or their generalized forms—dependent on transforms at the stage boundaries. When these weights depend on transforms from only one boundary, we call it *partial incompatibility*. For UPMs with partially incompatible stages, the probability of an attacker accessing all stages within $T$ time steps is $\left((T-2)\times(3pR)^2\right)^{S-1}$ (derived in Section C). Although this offers stronger protection than having no unextractability, it remains relatively high in practical settings.

Ideally, we would have *full incompatibility*, where at least one weight in each stage depends on the transforms from both boundaries. Skip connections inherently achieve this; thus transformer layers naturally satisfy full incompatibility (see Figure 2 (C), where both transforms are applied to $Q$ at each skip connection). For UPMs with fully incompatible stages, the approximate probability is now $(T-2)(3pR)^S$ (see Section C), which is very low for realistic settings (provided $R$ remains small).

## 3.2 Transforms

Transforms must be sufficiently random to resist brute-force or bias-exploiting attacks, strongly alter weights to prevent easy inversion, and remain numerically stable with minimal floating-point error. We find that these properties largely depend on the transforms' singular values (detailed in Appendix E). In particular, controlling the condition number is crucial, since numerical error increases exponentially with it. Thus, we select two transform classes:

**Haar orthogonal matrices.** Orthogonal matrices ($QQ^T = I$) have unit singular values, minimal numerical error, determinant $\pm 1$, and an easily computable inverse ($T^{-1} = T^T$). Uniform sampling from the Haar distribution ensures no exploitable bias.

**Low-condition number matrices.** To introduce higher frequencies while maintaining numerical stability, we use matrices of the form $UDV^T$, where $U, V$ are Haar orthogonal matrices, and $D = \text{diag}(se^{u_i})$, $u_i \sim \mathcal{U}[-\varepsilon, \varepsilon]$. Singular values now lie within $[se^{-\varepsilon}, se^{\varepsilon}]$, bounding condition number by $e^{2\varepsilon} \approx 1 + 2\varepsilon$. We cycle scale factor $s$ to prevent cumulative scale drift.

## 3.3 Training

While forward-pass equivalence under transforms is straightforward, ensuring correctness in the backward pass requires extra care. Transforms affect weights and gradients differently: if weights transform as $W_{(T)} = WT$ then its corresponding gradients satisfy $G_{(T)} = GT^{-T}$. For optimizers using only the first gradient moment $M^{(t)}$, we change the update step from $W^{(t+1)} = W^{(t)} - \eta f(G^{(t)}, M^{(t)})$ to $W_{(T)}^{(t+1)} = W^{(t)}T - \eta f\left(G^{(t)}T^{-T}, M^{(t)}T^{-T}\right)$. Furthermore, for some optimizers this is equivalent to update steps with the untransformed model when using orthogonal transforms: $W_{(T)}^{(t+1)} = \left(W^{(t)} - \eta f(G^{(t)}, M^{(t)})\right)_{(T)}$. We implement training with Muon [19], which satisfies both properties. Further details and derivations appear in Section F.

## 3.4 Inference

Although morphing steps leave the forward function unchanged mathematically, repeated transform folding accumulates numerical precision error. During training, gradient descent corrects such errors, but at inference there is no correction mechanism. We mitigate this by storing high-precision weights on disk and keeping a low-precision copy in GPU VRAM for inference. At each transform step, we apply the transform to the high-precision weights on disk, cast them to low precision, and reload them into GPU VRAM. Since transforms occur infrequently, disk-to-VRAM transfer overhead is negligible. Floating-point error analysis in Section E.2 confirms this effectively limits error accumulation.

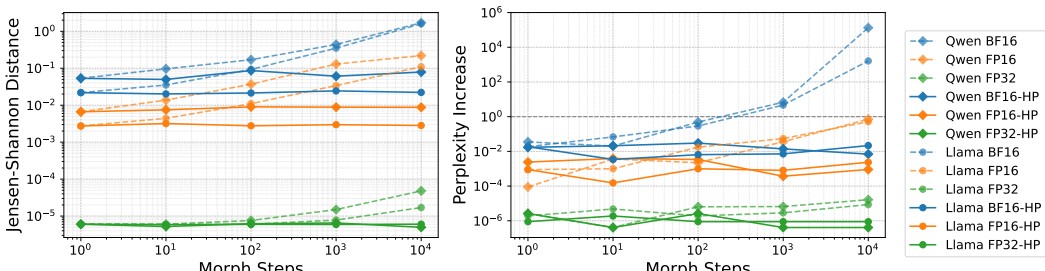

Figure 3: **Functional Equivalence during Inference.** We evaluate the cumulative effect of transforms by measuring logit drift (via Jensen–Shannon distance) and perplexity increase on the WikiText test split across different numbers of orthogonal morphing steps. Results are shown for Qwen 2.5 0.5B and Llama 3.2 1B across multiple precisions. Solid lines denote our high-precision (HP) workaround described in Section 3.4. Without the workaround, low precision leads to logit drift and rising perplexity, whereas with it, both remain stable over 10k steps.

# 4 Related Work

## 4.1 Decentralized and Federated Learning

Collaborative training primarily uses Data Parallelism (DP), where nodes store full model replicas, limiting scalability and exposing models to extraction [11, 32, 25, 9]. Alternative methods like SWARM [51] and Tasklets [66] shard models across nodes, and Hivemind [28] and Petals [6] futher improve decentralized inference with throughput reaching 6 tokens/sec for a 70B model. However, these methods have no mechanism to prevent model extraction. Federated Learning (FL) [30, 36, 20] keeps data decentralized, emphasizing privacy, and has been extended to both large models [65, 69] and incentives [21, 43]. Unlike FL, our work targets *weight secrecy* and robustness against untrusted participants, addressing fundamentally different challenges.

## 4.2 Communication Efficient Decentralized Training

Since decentralized training operates over low-bandwidth and high-latency networks, communication efficient methods that preserve training performance are essential. Many methods focus on the data parallel (DP) setting, where weight gradients need to be synchronized between devices [61, 7, 5, 11, 68]. However, they are insufficient for the pipeline parallel setup we consider, where activations and their gradients are also communicated between devices. Recently, several approaches have emerged to address this gap. Ajanthan et al. [2] further improves approaches that use pipeline schedule strategies to mask communication overhead by making optimization asynchronous. Ramasinghe et al. [47] losslessly compress activations and their gradients, and has been deployed in a public decentralized run [4]. As a result, pipeline parallel decentralized training is now well established.

## 4.3 Model Extraction Attacks

Black-box extraction attacks aim to replicate model capabilities without direct access. Tramèr et al. [59] demonstrated feasibility through prediction APIs, and Krishna et al. [27] extended this to neural language models via transfer learning and targeted queries. Wallace et al. [62] later extracted translation models via imitation learning, and Lee et al. [29] recently showed projection layers could be stolen with limited API queries. In contrast, we address extraction threats in collaborative training, where attackers participate directly rather than through restricted black-box access.

# 5 Results

We empirically validate our approach as follows. First, we verify that inference-time morphing preserves model behavior (Section 5.1), and quantify its practical overhead (Section 5.2). Next, we demonstrate that training under our framework matches unconstrained baselines for orthogonal transforms, consistent with theory (Section 5.3). Finally, we evaluate model-stitching and learning-based attacks (Section 5.4).

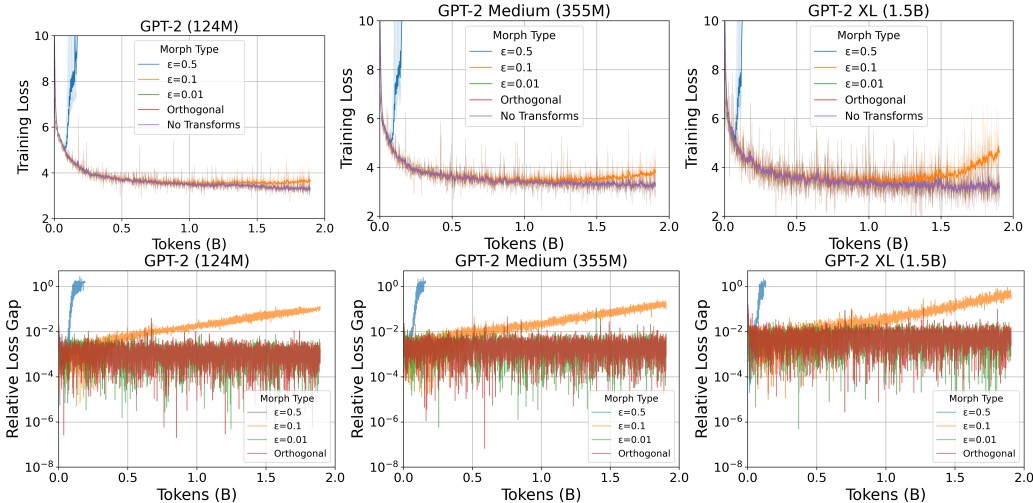

Figure 4: **UPM training.** Training on 1.9B FineWeb tokens with Muon across three GPT-2 scales, comparing our UPM framework (with various transforms, morphing occurs every 0.5M tokens) to a no-transform baseline. **Top:** training loss. **Bottom:** relative loss gap vs. the baseline. As per the analysis in Section 3.3, orthogonal transforms are behaviorally indistinguishable from the baseline (overlapping curves, constant and near-zero relative difference). A very small $\varepsilon$ ($\varepsilon = 0.01$) also tracks the baseline, $\varepsilon = 0.1$ drifts slowly and $\varepsilon = 0.5$ drifts slowly for at least 100 morphing steps before it becomes unstable and fails.

## 5.1 Functional Equivalence

While UPMs preserve the model's end-to-end function mathematically, during inference the precision error builds as discussed in Section 3.4. To quantify this, we start with two pretrained open weight LLMs, Qwen 2.5-0.5B [45] and Llama 3.2-1B [13], and iteratively apply our morphing steps. After each morphing step, we measure deviation from the original model in two areas: output logits distribution using Jensen-Shannon distance, and language modeling performance, using validation perplexity. Evaluation is performed on the WikiText (v2-raw) validation set [37]. We test three floating-point precisions: FP32 (single), FP16 (half), and BF16. We also apply the high-precision accumulation workaround from Section 3.4, where transforms are folded into high-precision weights on disk before downcasting to the GPU.

As shown in Figure 3, without the workaround (dashed lines) logit drift and perplexity grow with the number of morphing steps: FP32 changes remain small even after 10k steps, but FP16 and BF16 drift substantially, eventually rendering the model unusable. With the workaround (solid lines), growth is negligible across all precisions, consistent with our floating-point error analysis in Section E.2.

## 5.2 Inference Overhead

We consider the case of Llama 3.2 1B (which has dimension $D = 2048$), with a batch size $B = 4$, sequence length $S = 1024$, and morphing every 30s. The latter implies that new participants need to wait 1 minute before getting assigned a stage so that participants cannot leave and join in adjacent time steps (this time would be required for authentication checks anyway).

**Time:** On an NVIDIA A100 GPU (FP32), a morphing step requires approximately $0.05\,\mathrm{s}$, 95% of which is orthogonal matrix generation in FP64, yielding an amortized latency overhead of about 3%. This overhead can be further reduced by overlapping local computations with communication.

**Communication overhead:** Inference steps require sending activations of shape $(B, S, D)$ while morphing steps require sending transforms of shape $(D, D)$, resulting in an amortized bandwidth overhead of $\approx 0.1\%$.

**Memory overhead:** Each layer stores four additional $D \times D$ matrices, increasing storage by roughly 20% per layer, or 10% extra GPU memory overall (around 1GB for typical inference workloads).

|            | GPT-2 | GPT-2 Med | GPT-2 XL |
|------------|-------|-----------|----------|
| **Original**   | 3.28  | 3.26      | 3.16     |
| **Orthogonal** | 3.27  | 3.26      | 3.17     |
| $\varepsilon = 0.01$ | 3.28  | 3.26      | 3.16     |
| $\varepsilon = 0.1$  | 3.66  | 3.74      | 4.48     |
| $\varepsilon = 0.5$  | 5.09  | 5.01      | 5.13     |

Table 1: **UPM Training.** Final validation loss for the experiment in Figure 4, demonstrating the same trend: orthogonal transforms have identical behaviour to the baseline, and drift increase with $\varepsilon$.

## 5.3 Training

As explained in Section 3.3, we use the Muon optimizer for our training experiments. Muon is notable for having powered numerous speedrun records on CIFAR-10 and NanoGPT [18]. To demonstrate that UPMs support effective training, we build on a NanoGPT speedrun record [48] whose setup most closely mirrors modern GPT architectures: a GPT-2 model (124 M parameters) equipped with RMS-Norm, RoPE, and ReLU$^2$ activations. The goal of the NanoGPT speedrun is to reach a validation loss of 3.28 on FineWeb in as few tokens as possible. The particular record we use achieves this target in just 1.9 billion tokens. We also demonstrate this with larger GPT-2 sizes.

We show training loss curves for both the original and with UPMs in Figure 4, experimenting with different transforms, and report the final validation loss in Table 1. Consistent with the theory in Section 3.3, orthogonal transforms yield identical loss curves to the baseline, while non-orthogonal transforms ($\varepsilon > 0$) introduce gradual drift that accumulates with $\varepsilon$. Applying a transform every 100 steps adds only 1.6% time overhead and <1% memory overhead. These overheads are far smaller than at inference since the backward pass dominates both time and memory costs.

## 5.4 Attacks

In Section 3.1, we explained that weights accessed across stages at different transform time steps are inconsistent, and that the bridge matrix can only be determined if the time steps are adjacent (which we prevent by adding a waiting window to rejoining). We now examine two strategies for recovering consistent weights from inconsistent snapshots, and show they are impractical.

### 5.4.1 Model stitching attacks

esides transforms and weights, participants can observe intermediate activations $X_i$. Suppose an attacker obtains stage $i$ and $i + 1$ at time $t$ and $t' > t + 1$, so they have $V_i(t), T_i(t), X_i(t)$ and $T_i(t')$, $X_i(t'), U_{i+1}(t')$. To make the weights compatible, they need the bridge matrix $\hat{T} = T_i(t+1)...T_i(t')$. However, if the same input was provided to the network in both time steps, then $X_i(t') = X_i(t)\hat{T}$, so $\hat{T}$ can be expressed as a solution to a matrix system (see Section G for a detailed explanation). Furthermore, if $X_i(t)$ is full rank, then $\hat{T}$ can be solved for by inverting $X_i(t)$, and if not, $\hat{T}$ might be able to be determined using least squares. Such an attack would reduce the excludability ratio of the model close to zero. Although the compute requirement of reaching each stage in the protocol may be costly, it is extremely small compared to training from scratch.

However, requiring identical inputs across the time steps the weights were collected at is impractical and easy to defend against. Since the attack hinges on there being equal activations $X_i$ at both times steps, Section G both explains how an attacker could try to force this, and why those attacks still remain impractical and easily mitigated.

### 5.4.2 Learning based attacks

Beyond attacks that directly try to align incompatible weights that have been stitched together, adversaries might use learning-based attacks, fine-tuning the stitched weights to reconstruct the original model. If fine-tuning requires little compute, the excludability ratio approaches zero, undermining the model's economic security. We evaluate such attacks on a pretrained Llama 3.2-1B model using the FineWeb dataset [42]. We use both the base Llama model trained by Meta (on a wide variety of data) and a model we trained from scratch on FineWeb. We then apply transforms to these model (treating each transformer layer as a stage) and stitch together weights from different

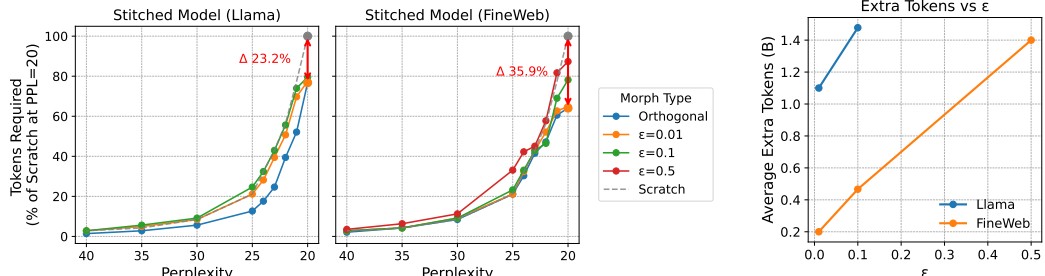

Figure 5: **Learning-based Attacks. Left:** Tokens required, as a percentage of training from scratch to a perplexity of 20, to reach various perplexities on FineWeb with a stitched Llama 3.2 1B model. We evaluate this with two sets of pretrained model weights that are morphed and then stitched: the base Llama model weights released by Meta (which is trained on a wide variety of data) and our model weights that we train from scratch on FineWeb. Transforms are applied such that the boundary between transformer layers are inconsistent. In all cases, finetuning the stitched model to a perplexity of 20 required **at least 60% of the compute** required to train the model from scratch to the same perplexity. **Right:** Average extra tokens in billions to reach a desired perplexity as $\varepsilon$ increases.

transforms. From this, we measure the compute required to reach various perplexity thresholds when finetuning (using FineWeb) compared to training from scratch. At the start of training we observe the perplexity of the stitched models are >60 000, comparable to a randomly initialized model. As shown in Figure 5, finetuning a stitched model that used orthogonal transforms still required roughly 60% of the compute of training from scratch. Small $\varepsilon$ transforms yield similar costs and larger $\varepsilon$ values slightly increase these costs, which we also show in terms of tokens.

Thus, UPMs are able to not only prevent direct weight set extraction, but also prevent information leakage that can allow an equivalent model to be produced with reasonable compute. Even when all weights are collected (though all are incompatible), the excludability ratio does not fall below 0.6. Such compute is not within the means of even medium sized attack coalitions.

## 6 Conclusion

We introduce Unextractable Protocol Models (UPMs), which apply random invertible transforms at each pipeline boundary to prevent piecewise Sybil attacks. In experiments with Qwen 2.5-0.5B and Llama 3.2-1B, during inference UPMs incur negligible logit and perplexity drift after 10 000 morphs, and add only $\sim 0.1\%$ bandwidth and $\sim 3\%$ latency overhead (amortized). We also demonstrated that training dynamics are unaffected for orthogonal transforms with the Muon optimizer, consistent with theory. Finally, we also show that stitching and learning-based attacks are computationally expensive and impractical. By binding model value to the protocol rather than static weights, UPMs enable compute providers to jointly serve and monetize large models without risking weight leakage. Our preliminary training results suggest a path toward fully trustless, unextractable training, positioning UPMs as a key technical pillar for open, decentralized AI.

**Limitations** Our security relies on standard cryptographic assumptions, notably that most participants are honest; thus, security weakens if a large portion is dishonest (see Section C). Our experimental evaluation is limited to single-machine simulations, excluding real-world decentralized implementations and side-channel attacks (e.g., timing, cache). Consequently, actual latency and bandwidth overhead depend on real deployment factors like batch sizes and network conditions.

**Broader Impact** This work develops Unextractable Protocol Models (UPMs) to allow for economic sustainability and open participation in large-scale decentralized AI. However, UPMs prevent the ability to inspect or disable model weights. This prevents safety evaluations and content-moderation controls, such as determining if activations have unwanted correlations, and could empower actors who wish to host unregulated or harmful AI services without fear of takedown or forensic analysis. To mitigate these, we suggest ensuring that diverse stakeholder participants jointly monitor model behavior and enforce usage policies, and continuously have transparent metrics evaluated and logged, enabling external safety validation.

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

# A   The economics of decentralized models

While the decentralization of large models allows the huge computation cost of training and serving inference at scale to be spread across a large number of participants, by nature such a setting is trustless. We can assume an honest majority of participants, but we must also assume that there are bad actors within the protocol. As a result, any information that participants receive unprotected access to, be it code, data, or weights, can be utilized outside the protocol. We now explain what must be protected in order for the economics of decentralized protocols for large models to be rational, and why unextractability is sufficient to guarantee this.

## A.1   Appropriability and Excludability

The key issue here is called the *appropriability problem* in economics: the difficulty for producers to appropriate (capture) value from their product [3, 56, 44]. Not only do they need to recoup the cost of production, but there needs to be surplus value as an incentive for the production. In the context of decentralized protocol models, the producers are the participants of the protocol and the product is the ability to utilize the model via the protocol (query the trained model). The value of this model utility (the product) is largely governed by the capability of the model.

Increasing the appropriability of the product is often discussed in terms of the related economic notion of *excludability*, the degree to which non-paying consumers can be denied access to the product [52, 39, 22]. By only allowing access to query the model via the protocol, the protocol can then monetize serving inference for general consumers and distribute returns to participants who supplied compute. This gives participants the incentive to supply compute in the first place. However, if one of the participants can gain access to the full model, then there is the ability to query the model outside the protocol and the excludability of the protocol disappears.

This is the motivation for unextractable protocol models, ensuring that no participant can recreate the full weight set while participating in the protocol, and thus guaranteeing the excludability of the protocol.

## A.2   Partial Excludability

Furthermore, we can consider *partial excludability*: if an attacker can produce an equivalent model using whatever knowledge they gained from participating in the protocol and additionally some amount of compute that is substantial (which includes the compute required to participate in the protocol), then the excludability is fractional and depends on how large that required compute cost is relative to the overall compute power needed to train the model from scratch.

Let us formalize partial excludability for our situation. Assume that a performance measure $\mathcal{P}(M)$ has been defined for any model $M$ along with a tolerance value $\varepsilon$, for example the validation perplexity of the model on the dataset it is being trained on with a tolerance of 0.5 perplexity, or perhaps the validation perplexity on an independent dataset. Let us also consider a compute measure $\mathcal{C}(M)$ for obtaining a model $M$, for example the amount of flops, GPU hours, or cost of compute.

Consider the case of an original model trained from scratch in the protocol $\widehat{M}$ with performance measure $\mathcal{P}(\widehat{M})$, and a model $M'$ that an attacker has derived from participating in the protocol with performance measure $\mathcal{P}(M')$, where $|\mathcal{P}(\widehat{M}) - \mathcal{P}(M')| \leq \varepsilon$. Then to quantify how excludable $\widehat{M}$ is and assuming this attack is the best possible attack, we define its *excludability ratio* as the ratio between the compute $\mathcal{C}(M')$ required to derive $M'$, and the compute $\mathcal{C}(\widehat{M})$ required to train $\widehat{M}$ from scratch

$$\mathcal{E}_r = \frac{\mathcal{C}(M')}{\mathcal{C}(\widehat{M})}. \tag{6}$$

# B   Possible Extraction Scenarios and their Excludability Ratio

Let us consider possible extraction scenarios and their excludability ratio, noting that we want to ensure that all feasible attacks have a high excludability ratio. In fact, an excludability ratio of more

than 0.1 is very likely to still be out of the reach of individual attackers / attack coalitions, as this is still an enormous amount of compute.

Since we can consider the case when the code and data are completely open (as it is economically feasible with our framework), with enough compute anyone can train their own model. However, this requires the same amount of compute as the decentralized protocol, $\mathcal{C}$, and so the excludability ratio is one. On the other hand, if an attacker can somehow compromise the security of the protocol and directly gain access to all trained weights, then their required compute is zero and the excludability ratio is also zero. This is extremely unlikely as we assume that standard security measures have been put in place.

Now let us consider the situation where the attacker participates in the protocol. If the attacker manages to collect weights from each stage over multiple time steps and there are no unextractability measures put in place, then the required compute is whatever was needed to participate in the protocol until the attacker collected all weights. While this compute might be substantial for an individual (as participating in the protocol requires compute), it is dwarfed by the enormous amount of compute needed for training the large model, so the excludability ratio is essentially zero.

On the other hand, let us say that there are still no unextractability measures in place, and the attacker manages to gain access to a large subset of the entire weight set. By keeping those weights fixed, and training all the other weights in the model, the attacker may be able to gain access to a model with equivalent capability. If the compute required for training is some fraction $\alpha$ of $\mathcal{C}$, then the excludability ratio is $\alpha$. Similarly, when there are unextractability measures in place, the attacker might be able to collect a full set of incompatible weights, and finetune from these weights in order to align them and thus gain a compatible model with some fraction $\alpha$ of $\mathcal{C}$. In both these cases, if $\alpha$ is small enough that an adversarial entity has access to $\alpha\mathcal{C}$ compute but not $\mathcal{C}$ compute (so cannot train the model themselves but could carry out one of these attacks), then the appropriability of the model is severely reduced.

Now let us consider attacks with our unextractability framework in place. It is still possible for an attacker to participate in the protocol and manage to collect weights from each stage over multiple time steps as well as the transforms needed to make the weights compatible. However, as shown in Section C, the probability of this happening can be made vanishingly small, especially for the case of transformers.

Another possible attack is to use intermediate results gained while in the protocol to solve for the required bridge matrices. We discuss such attacks in Sections 5.4.1 and G, including easy ways to prevent them.

Finally, an attacker might use whatever set of weights they obtained while participating in the protocol, which will be incompatible with each other, to train an equivalent model with less compute than training from scratch. This is the learning based attack discussed in Section 5.4.2. We demonstrated that even with access to a full set of (incompatible) weights, the excludability ratio remains quite high.

## C   Decentralized Model Pipeline Setup and Extraction Probabilities

Let us first summarize our setting from Section 2.

- We consider the situation of a pipeline with $S$ stages and $R$ replicas of each stage, and each of the $RS$ slots (individual replicas of a stage) require a compute node.
- The combined compute pool from all participants is $C > RS$ nodes, nodes are allocated to slots at random (hiding the stage identity)
- The protocol might implement stricter measures like the nodes belonging to a single participant can only be allocated to slots in a single stage
- We assume that the total nodes an attacker controls is $A \ll RS < C$. This is reasonable since we are considering a large-scale model that requires collaboration from multiple participants, meaning no one participant has the computational capability to hold all the nodes in the protocol. Thus the fraction that the attacker controls is $p = \frac{A}{C} \ll 1$.

For further clarity, we assume the following standard requirements of the protocol:

- In order to participate, a participant has to prove they have at least one compute node (i.e., they have the GPU compute to at least run a single stage)
- Secure, authenticated communication is established between nodes.
- Decentralized work verification ensures that all participants contribute valid activations.

Finally, in order to compute the probability of an attacker being able to extract the entire model using a piecewise Sybil attack in various scenarios, let us assume that $RS$ is small compared to $C$. This is not unreasonable since the monetary worth of large models attracts participants (c.f. blockchain). The benefit of this is that choosing compute to fill $RS$ slots from $C$ nodes is well approximated by sampling with replacement.

Thus the probability of a particular slot being controlled by an attacker is then $p$, the expected number of attacker-occupied slots per timestep is $pRS$, and the expected number across $T$ timesteps is $pRST$. Furthermore the probability of the attacker getting access to a particular stage $i$ (in any replica) in a time step is

$$p_i = 1 - (1 - p)^R. \tag{7}$$

We now calculate the probability of an attacker being successful under different scenarios. In many cases we use a modification of the binomial approximation: $1 - (1 - x)^\alpha \approx \alpha x$ for $|x| < 1, |\alpha x| \ll 1$. Note that this is an over-approximation for $x > 0$ (which is the region we use).

**Default setup (no unextractability measures).** In the default case, for the attacker to be successful the attacker must have access to all $S$ stages, each of which can be from any replica and from any time step. Let us assume that the attacker is in the protocol for $T$ time steps, then they can form a complete weight set by getting access to the stages over the $T$ time steps.

The probability of ever capturing stage $i$ within $T$ time steps is $p_{i,T} = 1 - (1 - p_i)^T = 1 - (1 - p)^{RT}$. This means that the probability of getting access to all $S$ stages over the $T$ time steps is

$$\left[1 - (1 - p)^{RT}\right]^S \approx (pRT)^S. \tag{8}$$

Note that $p, R, S$ are fixed with $p$ being very small ($10^{-5} - 10^{-2}$), and $R$ and $S$ being small integers ($R$ is probably 3-100, $S$ is around 10-100). However $T$ can be very large (millions or billions of steps) so the probability is almost certain for the attacker (in practice there would be some mechanisms to prevent easy switching of stages between timesteps, slightly decreasing the chance).

## C.1  With Unextractability

Now we consider the case when we make the stages time-varying, so that different stages are not compatible across time steps. Note however, that the bridge matrix between neighboring stages is available for subsequent time steps (see Section 3.1). Thus we consider two cases

- The weights in any stage $i$ can be partitioned into two sets, $\theta_{i,1}$ and $\theta_{i,2}$, where the transforms between stage $i - 1$ and $i$ are only applied to weights in $\theta_{i,1}$, and the transforms between stage $i$ and $i + 1$ are only applied to weights in $\theta_{i,2}$. We call this **partial incompatibility**.
- The weights in any stage $i$ cannot be partitioned in the above manner, thus there is at least one weight in the stage that depends on both the transform between stage $i - 1$ and $i$ and also depends on the transform between stage $i$ and $i + 1$. We call this **full incompatibility**.

**Full Incompatibility.** Let us first consider full incompatibility. Then if an attacker gets access to stage $i$ in time step $t$, in order for it to be compatible the weights of stage $i - 1$ and stage $i + 1$ need to be also gained at time $t$. Thus between any set of three consecutive stages, the weight of all three stages have to be at the same transform time step. Since participants only have the bridge matrix (to convert weights from one time-step to another) between subsequent time steps, this can be done if stage $i - 1$ and $i + 1$ are one time step away from stage $i$. However since this needs to hold between any set of three consecutive stages, this can only work if all stages have a way to convert their weights to a single time step.

Thus the attacker needs **all stages within three consecutive time steps**, as then any stage's weights can be converted to the middle time step. Then the probability of a specific stage being covered

within a specific three time steps is $p_{i,3} = 1 - (1-p)^{3R}$, and the probability of all $S$ stages covered in the three time steps is $p_{i,3}^S = (1 - (1-p)^{3R})^S$. In $T$ time steps there are $T-2$ sets of three consecutive time steps, so the probability of a successful attack over $T$ time stages is

$$1 - (1 - p_{i,3}^S)^{T-2} = 1 - (1 - (1 - (1-p)^{3R})^S)^{T-2} \approx (T-2)(3pR)^S. \tag{9}$$

Thus as long as $3pR$ is reasonably small, e.g. less than 0.2, the probability of success is small even for very large $T$.

**Partial Incompatibility.** Now let us consider partial incompatibility. Then each pair of $\theta_{i,2}$, $\theta_{i+1,1}$ between stages $i$ and $i+1$ must be at the same time-step, though any other pair can be at different time steps. Furthermore, since participants have bridge matrices between subsequent time steps, these can be transformed to neighboring time steps only.

Thus the attacker needs **each pair of stages $i$ and $i+1$ within three consecutive time steps**, as then the boundary weights of those two stages can be converted to the middle time step, and all such pairing can be stitched together. Thus the probability of getting two specific stages within three time steps is $p_{i,3}^2$, so getting this within the $T-2$ opportunities is

$$1 - (1 - p_{i,3}^2)^{T-2} \approx (T-2)(3pR)^2 \tag{10}$$

and finally doing this for all $S-1$ pairs of stage boundaries is

$$(1 - (1 - p_{i,3}^2)^{T-2})^{S-1} \approx ((T-2)(3pR)^2)^{S-1}. \tag{11}$$

Note that $T$ is now scaled by $S$, unlike in the full incompatibility case. While this is much less probable than the default setup, with reasonable $p$ there is a high probability for large $T$. However, unlike the default setup, the requirements here are much easier to defend against, as shown in the next section.

**Achieving Full Incompatibility.** An easy way to achieve full incompatibility is to have skip connections in the stage. Then the stage is of the form $\Phi_i(XU_i)V_i + XQ_i$, and transforms get applied as $\Phi_i(XT_{i-1}U_i)V_iT_i + XT_{i-1}Q_iT_i$, thus making $Q_i(t) = T_{i-1}(t)Q_i(t-1)T_i(t)$ be dependent on both neighbors.

### C.2 Mechanisms

A key issue is therefore to reduce the chance that an attacker can control two neighboring stages within three consecutive time steps. We can therefore take some effective measure

- Once compute has been assigned to a stage, have the compute keep that stage (note that there are other replicas for redundancy and for checking malicious activity)

- To stop attacker compute from switching (in the hopes of using the same compute to get a stage they need) we require newly joined nodes to wait two time steps before participating.

- Have a preference for assigning nodes that have been waiting for longer to join the pipeline.

Thus, an attacker can no longer switch compute within the required three consecutive time steps. Instead, assuming they have $A > 2$ nodes, they need to switch all $A$ nodes at the same time and hope that at least two of the nodes get allocated to the required two neighboring stages within three time steps. However, assuming node dropout is not very often, this is highly unlikely: if there is a lot of nodes waiting most likely other nodes will get assigned, and if there is not a lot of nodes waiting then the attacker nodes (even when masquerading as new identities) are likely to be assigned back their original slots.

Finally, note that inference is the hardest case, as weights never go "stale", whereas during the early part of training, weights will drastically change over time and thus go stale. Thus, there is an extra time requirement on the attacker during (early stage) training.

# D Applying UPMs to different components and architectures

## D.1 MLPs

Let us consider an $n$-layer MLP $g_n \circ g_{n-1} \circ ...g_1$ where $g_i(X) = \sigma\left(XU_i\right)V_i$, the $U_i$ are weight matrices, $V_i = I$, and $\sigma$ is an element-wise activation function. Let the output after the $i^{\text{th}}$ layer be $X_i$, i.e. $X_i = (g_i \circ g_{i-1} \circ ...g_1)(X_0)$.

When we apply transforms, we have that

$$U_i(t) = T_{i-1}^{-1}(t)U_i(t-1) = T_{i-1}^{-1}(t)...T_{i-1}^{-1}(1)U_i \tag{12}$$

$$V_i(t) = V_i(t-1)T_i(t) = IT_i(1)...T_i(t) \tag{13}$$

$$X_i(t) = X_i(t-1)T_i(t) = X_iT_i(1)...T_i(t). \tag{14}$$

Note that we only have **partial incompatibility** (no matrices in a stage depend on both boundaries).

**Using weights at different time steps**  Let us assume an attacker tries to steal a consistent set of weights by having different blocks in different time steps. In particular, let us assume they have $f_i$ at time $t_1$ and $f_{i+1}$ at time $t_2 > t_1$. Then they have

$$U_i(t_1) = T_{i-1}^{-1}(t_1)U_i(t_1-1) = T_{i-1}^{-1}(t_1)...T_{i-1}^{-1}(1)U_i \tag{15}$$

$$V_i(t_1) = V_i(t_1-1)T_i(t_1) = IT_i(1)...T_i(t_1) \tag{16}$$

$$X_i(t_1) = X_i(t_1-1)T_i(t_1) = X_iT_i(1)...T_i(t_1) \tag{17}$$

$$U_{i+1}(t_2) = T_i^{-1}(t_2)U_{i+1}(t_2-1) = T_i^{-1}(t_2)...T_i^{-1}(1)U_{i+1} \tag{18}$$

$$V_{i+1}(t_2) = V_{i+1}(t_2-1)T_{i+1}(t_2) = IT_{i+1}(1)...T_{i+1}(t_2) \tag{19}$$

$$X_{i+1}(t_2) = X_{i+1}(t_2-1)T_{i+1}(t_2) = X_{i+1}T_{i+1}(1)...T_{i+1}(t_2) \tag{20}$$

as well as $T_{i-1}(t_1), T_i(t_1), T_i(t_2), T_{i+1}(t_2)$.

In order to get consistent weights, they need to have $V_i(t), U_{i+1}(t)$ for the same time step $t$. Thus they need one of the following configurations

- $\mathbf{V_i(0)} = \mathbf{I}$, $U_{i+1}(0) = U_{i+1} = \mathbf{V_i(t_1)}\hat{T}\mathbf{U_{i+1}(t_2)}$

- $\mathbf{V_i(t_1)}$, $U_{i+1}(t_1) = \hat{T}\mathbf{U_{i+1}(t_2)}$

- $V_i(t_2) = \mathbf{V_i(t_1)}\hat{T}$, $\mathbf{U_{i+1}(t_2)}$

where $\hat{T} = T_i(t_1+1)...T_i(t_2-1)\mathbf{T_i(t_2)}$ is the bridge matrix, and anything the attacker has is bolded. Note that the only thing missing in all three configurations is exactly $\hat{T}$, which the attacker only knows if $t_2 = t_1 + 1$. Otherwise we need to factorize $U_{i+1}(t_2)$ into $\hat{T}^{-1}V_i(t_1)^{-1}U_{i+1}$ where only $V_i(t_1)^{-1}$ is known and we know that $V_i(t_1)^{-1}$ and $\hat{T}$ are square and invertible. However there are infinitely many possibilities, as $\hat{T}V^{-1}U = \left(\hat{T}A\right)V^{-1}\left(VA^{-1}V^{-1}U\right) = \hat{T}'V^{-1}U'$ for any invertible matrix $A$.

## D.2 RMSNorm

As explained in the main paper, we want RMSNorm applied layers to follow

$$\Phi\left(\text{RMSNorm}\left(XT\right)U_i\right)V_i = \Phi\left(\text{RMSNorm}\left(X_{i-1}\right)TU_i\right)V_i \tag{21}$$

and in order to do that we (1) introduce a transform accumulation matrix into the normalization: $\text{norm}(X) \rightarrow \text{norm}(XQ(t))$, where $Q(0)$ is initialized to an orthogonal matrix, and (2) move the feature scaling weights $\text{diag}(w)$ into the next layer weights $U_i(0) = \text{diag}(w)U_i$ (which for training is equivalent to removing them). Then our layer is now of the form

$$f_i\left(X_{i-1}\right) = \Phi\left(\text{norm}\left(X_{i-1}Q_i\right)X_{i-1}U_i\right)V_i, \tag{22}$$

and so after a morphing step $Q_i \mapsto T^{-1}Q_i$ we have the desired functional equivalence

$$f_i\left(X_{i-1}T\right) = \Phi\left(\text{norm}\left(X_{i-1}TT^{-1}Q_i\right)X_{i-1}TU_i\right)V_i \tag{23}$$

$$= \Phi\left(\text{norm}\left(X_{i-1}Q_i\right)X_{i-1}TU_i\right)V_i \tag{24}$$

$$= \Phi\left(\text{RMSNorm}(X_{i-1})TU_i\right)V_i. \tag{25}$$

where the RMSNorm in the last line is the original version with $\text{diag}(w)$ set to $I$ (since we have already moved $\text{diag}(w)$ into $U_i(t)$).

### D.3 Transformer

Let us consider a modern transformer layer with RMSNorm prenormalization. After applying the above change for RMSNorm layers, we have that

$$\text{RMSNorm}(X) = \text{norm}(XQ)X \tag{26}$$

where $Q$ is a fixed (random) orthogonal matrix and the weights have been subsumed into the next layer.

Thus the transformer layer is given by

$$f_i(X_i) = (\text{MLP} \circ \text{ATT})(X_i) \tag{27}$$

$$\text{ATT}(X) = \text{SA}(\text{RMSNorm}_1(X)) + X \tag{28}$$

$$\text{MLP}(X) = \sigma(\text{RMSNorm}_2(X)U_{w1})V_{w2} + X \tag{29}$$

where

$$\text{SA}(X) = \text{Att}(XU_k, XU_q, XU_v)V_o \tag{30}$$

$$\text{RMSNorm}_1(X) = \text{norm}(XQ_{n1})X \tag{31}$$

$$\text{RMSNorm}_2(X) = \text{norm}(XQ_{n2})X. \tag{32}$$

Applying transforms (see Figure 2 (C)) we get

$$\text{ATT}(X) = \text{SA}(\text{RMSNorm}_1(X)) + XQ_1(t) \tag{33}$$

$$\text{MLP}(X) = \sigma(\text{RMSNorm}_2(X)U_{w1}(t))V_{w2}(t) + XQ_2(t) \tag{34}$$

$$\text{SA}(X) = \text{Att}(XU_k(t), XU_q(t), XU_v(t))V_o(t) \tag{35}$$

$$\text{RMSNorm}_1(X) = \text{norm}(XQ_{n1}(t))X \tag{36}$$

$$\text{RMSNorm}_2(X) = \text{norm}(XQ_{n2}(t))X \tag{37}$$

where if the incoming transform is $T_{i-1}$ and the outgoing transform is $T_i$:

$$Q_{n1}(t) = T_{i-1}^{-1}(t)Q_{n1}(t-1) \tag{38}$$

$$U_k(t) = T_{i-1}^{-1}(t)U_k(t-1) \tag{39}$$

$$U_q(t) = T_{i-1}^{-1}(t)U_q(t-1) \tag{40}$$

$$U_v(t) = T_{i-1}^{-1}(t)U_v(t-1) \tag{41}$$

$$V_o(t) = V_o(t-1)T_{int}(t) \tag{42}$$

$$Q_1(t) = T_{i-1}^{-1}(t)Q_1(t-1)T_{int}(t) \tag{43}$$

$$Q_{n2}(t) = T_{int}^{-1}(t)Q_{n2}(t-1) \tag{44}$$

$$U_{w1}(t) = T_{int}^{-1}(t)U_{w1}(t-1) \tag{45}$$

$$V_{w2}(t) = V_{w2}(t-1)T_i(t) \tag{46}$$

$$Q_2(t) = T_{int}^{-1}(t)Q_2(t-1)T_i(t). \tag{47}$$

Here we have added an intermediate transform $T_{int}$, which is not redundant due to $Q_1$ and $Q_2$.

Note that we have **full incompatibility** due to the skip connections: $Q_1$ depends on $T_{i-1}$ and $T_{int}$ and $Q_2$ depends on $T_{int}$ and $T_i$, so there is no way to partition the matrices in a way that one partition can only depend on $T_i$ without $T_{i-1}$ and vice-versa for the other.

For SwiGLU MLPs, we have that

$$\text{MLP}(X) = \left(\text{SiLU}(\text{RMSNorm}_2(X)U_{w1}) \cdot (\text{RMSNorm}_2(X)U_{w3})\right) V_{w2} + X \tag{48}$$

$$\tag{49}$$

so we have the additional transformed weight

$$U_{w3}(t) = T_{int}^{-1}(t)U_{w3}(t-1). \tag{50}$$

# E  Transforms

We first analyze the key properties we want in our transforms, then describe the specific families we use.

## E.1  Transform Properties

There are multiple factors that we would like in order for the weights to be sufficiently scrambled:

- the scrambling is done sufficiently randomly with respect to an infinite class of transforms, so it is not possible to brute force all possible transforms or exploit there being a bias towards certain transforms within the class
- the new weights are a far distance away from the old weights (so the effect on the network of a single transform without its inverse is not small)
- the transforms introduce high frequencies, which makes finetuning transformed parameters difficult.

On the other hand, we want to control the transforms such that

- the floating point error is not too large
- the result does not overflow or underflow.

It turns out that we can link a lot of these properties to the singular values of our transforms, which we will now define notation for. For a matrix $M \in \mathbb{R}^{d \times d}$ we will denote its singular values by $\{\sigma_i(M)\}_{i=1}^d$ where the indexing orders them from largest to smallest. Some important properties are: the condition number is given by $\kappa(M) = \frac{\sigma_1(M)}{\sigma_d(M)}$, the determinant is given by $\det(M) = \prod_{i=1}^d \sigma_i(M)$, and the Frobenius norm is given by $\|M\|_F = \sqrt{\sum_{i=1}^d \sigma_i(M)^2}$.

**Floating point error**  The floating point error (relative forward error) of computing $W' = WT_1....T_n$ by post-multiplying the $n$ transforms $T_i$ one at a time is (see Section E.2 for proof)

$$\frac{\|\hat{W}' - W'\|_2}{\|W'\|_2} \lesssim \begin{cases} \gamma_d \kappa_W \frac{\kappa(\kappa^n - 1)}{\kappa - 1} & \kappa \neq 1 \\ n\gamma_d \kappa_W & \kappa = 1 \end{cases} \tag{51}$$

where $W$ and the $T_i$ are $d \times d$, $W$ has a condition number of $\kappa_W$ and the $T_i$ have a condition number of $\kappa$, and $\gamma_d = \frac{du}{1-du}$ where $u$ is the unit roundoff (half machine epsilon). Thus if $\kappa \neq 1$ then the error grows exponentially in the number of transforms, which severely limits the number of transforms that can be applied before the error is too large. Thus the singular values of the transforms should be as close to each other as possible.

**Scale Control**  When applying morphing steps $V \mapsto VT$, $U \mapsto T^{-1}U$, the resulting size of the weights as measured by the Frobenius norm is governed by the spectra of $T$. The von-Neumann trace-inequality [16] gives

$$\|V T\|_F \leq \sqrt{\sum_{i=1}^d \left[\sigma_i(V)\,\sigma_i(T)\right]^2}, \qquad \|T^{-1}U\|_F \leq \sqrt{\sum_{i=1}^d \left[\sigma_i(U)\sigma_i(T)^{-1}\right]^2}. \tag{52}$$

Thus if the singular values of $T$ are all large (small), then the Frobenius norm of $V$ greatly increases (decreases) and Frobenius norm of $U$ greatly decreases (increases). On the other hand, if $T$ has both large and small singular values, then the effect on $U$ and $V$ is unpredictable, and likely to make both

grow (note that just one singular value needs to blow up). However from the previous paragraph we want to keep the condition number small, so we anyway keep the singular values of $T$ close to each other. Thus to control the scale of the weights, we either have the singular values close to one, or cycle between large and small singular values each time step so that the scaling cancels out on every second time step.

**Introducing high frequencies.** It has been observed that neural networks trained with SGD are biased towards low frequency functions, in particular the low frequencies of the target function are learned before the high frequencies [64, 63, 46]. Thus we want to add transforms into weights such that just applying the transform without its inverse introduces high frequencies, meaning to undo the transform with gradient descent requires learning to cancel out the introduced high frequencies (a high frequency target). To inject high-frequency behavior into the network's input–output map, our transforms should amplify the directions in weight-space that correspond to high-frequency components of the learned function without amplifying other directions, i.e. have larger singular values for high-frequency directions and smaller ones for the low-frequency directions. With our random transforms, a simple approach is to sample a random orthonormal basis (singular vectors) and assign a spectrum of singular values that is skewed toward larger values, on average this will inject more high-frequency content than a flat spectrum would.

**Scrambling distance.** We need to generate transforms from a large class of transforms randomly, and ideally uniform randomly so that there is no bias towards specific transforms in that class. However most classes of transforms will contain transforms that do not do a good job at scrambling: e.g. the class of orthogonal matrices contains rotation matrices, and if $T$ corresponds to the rotation matrix of a small angle then $\|W - TW\|_2$ might be small. Furthermore, folding in multiple rotation matrices might make them cancel out. In order for an attacker to not be able to align weights from different time steps, we need to ensure that transforms always drastically change the weight matrix and never cycle back close to the original weights. Thus we need to ensure that our class of transforms is distributed enough that low scrambling matrices have an extremely low probability.

## E.2 Controlling Floating Point Error Drift

As discussed in Section 3.4, when folding in transforms a large number of times the floating point error builds. We now analyze this formally.

### E.2.1 FP analysis

We use the standard error bound for matrix multiplication from Higham [15]. Given two $d \times d$ matrices $A$ and $B$ in precision $P$ their error is bounded by

$$\|fl(AB) - AB\|_2 \leq \gamma_d \|A\|_2 \|B\|_2. \tag{53}$$

where $fl(\cdot)$ is floating point multiplication in precision $P$, $\gamma_d = \frac{du_P}{1 - du_P} \approx du_P$ and $u_P$ is the unit roundoff in precision $P$.

Given some computation $C$ whose result in floating point multiplication in precision $P$ is denoted $C^{(P)}$, we want to compute relative errors

$$\frac{\|C^{(P)} - C\|_2}{\|C\|_2} \tag{54}$$

with respect to the condition number of the matrices.

Some important results are: $\|AB\|_2 \leq \|A\|_2 \|B\|_2$, $\|A\|_2 = \sigma_{\max}(A)$, $\|AB\|_2 \geq \sigma_{\min}(A)\sigma_{\min}(B)$, $\kappa(A) = \frac{\sigma_{\max}(A)}{\sigma_{\min}(A)}$, $\kappa(AB) \leq \kappa(A)\kappa(B)$ and $\|AB\|_2 \geq \sigma_{\min}(A)\|B\|_2$

**Single Product:** The relative error in terms of the condition number of the matrices for a single matrix multiply is:

$$\frac{\|fl(AB) - AB\|_2}{\|AB\|_2} \leq \gamma_d \frac{\sigma_{\max}(A)\sigma_{\max}(B)}{\sigma_{\min}(A)\sigma_{\min}(B)} \tag{55}$$

$$= \gamma_d \kappa(A)\kappa(B). \tag{56}$$

**Chain of Products:** Now let us consider doing a chain of $n+1$ matrices, e.g., $n$ transforms $T_i$ applied to an initial weight matrix $W$: $C_n = WT_1...T_n$. Let us denote the floating point representation by $C^{(P)} = fl(fl(WT_1)...T_n)$.

Now the error in the $k$th multiplication is bounded by

$$\|fl(C_{k-1}T_k) - C_k\|_2 \leq \gamma_d \|C_{k-1}\|_2 \|T_k\|_2. \tag{57}$$

We will bound the total error in the chain as the sum of the errors in each multiplication times the norm of the remaining computation done to that multiplication result (first order approximation neglecting quadratic terms):

$$\|C_n^{(P)} - C_n\|_2 \approx \sum_{k=1}^{n} \|fl(C_{k-1}T_k) - C_k\|_2 \|T_{k+1}...T_n\|_2 \tag{58}$$

$$\leq \gamma_d \sum_{k=1}^{n} \|C_{k-1}\|_2 \|T_k\|_2 \|T_{k+1}...T_n\|_2 \tag{59}$$

so the total relative error is approximately bounded by

$$\frac{\|C_n^{(P)} - C_n\|_2}{\|C_n\|_2} \lesssim \gamma_d \sum_{k=1}^{n} \frac{\|C_{k-1}\|_2 \|T_k\|_2 \|T_{k+1}...T_n\|_2}{\|C_n\|_2} \tag{60}$$

$$\leq \gamma_d \sum_{k=1}^{n} \frac{\|C_{k-1}\|_2 \|T_k\|_2 \|T_{k+1}...T_n\|_2}{\sigma_{\min}(C_{k-1}T_k)\|T_{k+1}...T_n\|_2} \tag{61}$$

$$\leq \gamma_d \sum_{k=1}^{n} \kappa(C_{k-1})\kappa(T_k) \tag{62}$$

$$\leq \gamma_d \kappa(W) \sum_{k=1}^{n} \prod_{i=1}^{k} \kappa(T_i). \tag{63}$$

If each $T_i$ have the same condition number $\kappa(T_i) = \kappa$, then this is a geometric sum given by

$$\frac{\|C_n^{(P)} - C_n\|_2}{\|C_n\|_2} \lesssim \begin{cases} \gamma_d \kappa_W \frac{\kappa(\kappa^n - 1)}{\kappa - 1} & \kappa \neq 1 \\ n\gamma_d \kappa_W & \kappa = 1 \end{cases}. \tag{64}$$

### E.2.2 Comparison for different precisions

Let us consider two $d \times d$ weight matrices, $W_1$ and $W_2$, which are stored on GPU in a low-precision datatype $LP$, e.g. FP32 or FP16. Their matrix multiplication in precision $LP$ is $W_1^{(LP)}W_2^{(LP)}$.

Note that when we generate an orthogonal transform $T(t)$, we usually do this in high precision $HP$ anyway (Haar orthogonal is very sensitive to precision), specifically FP64. Thus, since the weight matrices on the GPU are in low precision $LP$, we have three options

- (A) Cast down transforms to $LP$ and then fold them into the weights
- (B) Whenever a transform needs to be folded in, cast the weights to $HP$, fold in the transforms, and then cast the result back to $LP$
- (C) Keep a copy of the weights in $HP$, and always fold transforms into this copy. Whenever the model needs to be queried, update the weight matrix by casting this copy to $LP$.

Let us now analyze the floating point error behavior of these three methods for a chain $WT_1...T_n$ where the $T_i$ are orthogonal (so $\kappa(T-i) = 1$). From above, we have that the relative error is bounded by $ndu_P\kappa_W$.

Now in practice for large models, $d$ is one of 1024, 2048, 4096 or 8192, so we will use $d = 10^3$. The condition number of a weight matrix $\kappa_W$ is usually around $10^2 - 10^5$, we will use $\kappa_W = 10^3$. For FP64, $u_{FP64} \approx 10^{-16}$, and for lower precisions we have $u_{FP32} \approx 10^{-8}$, $u_{FP16} \approx 10^{-4}$ and $u_{BF16} \approx 10^{-3}$. Then the floating point error in a single multiply is given by $10^{-10}$ for FP64, $10^{-2}$ for FP32, $10^2$ for FP16 and $10^3$ for BF16.

Now let us consider our three options from before with $n = 10^4$ transforms applied and low precision set as FP32. Then the relative error is

- (A): $ndu_P\kappa_W = 10^4 10^3 10^{-8} 10^3 = 10^2$

- (B): a single matrix multiplication becomes $du_{HP}\kappa(A)\kappa(B) + u_{LP}$, however this then accumulates in subsequent matrix multiplications so we can think of the bound as essentially something like $ndu_{P'}\kappa_W$ where $u_{P'}$ is somewhere between $u_{HP}$ and $u_{LP}$. Thus overall still bounded by $ndu_{LP}\kappa(W) = 10^4 10^3 10^{-8} 10^3 = 10^2$

- (C): this is the error in high precision plus a single low precision transfer, so $ndu_{HP}\kappa_W + u_{LP} = 10^4 10^3 10^{-16} 10^3 + 10^{-8} = 10^{-6}$.

### E.3 Transform Classes

**Haar distributed orthogonal matrices.** The first class is orthogonal matrices, which by definition satisfy $QQ^T = Q^T Q = I$. Their singular values are all one, which leads to nice properties for our use case: they have a determinant of 1 or -1 (so they keep the relative scale of the weights) and they have a condition number of one (so they do not introduce too much floating error). Furthermore $T^{-1} = T^T$ so the inverse is both easy to compute and does not require additional storage. We can also generate them uniform randomly from the Haar distribution over the set of orthogonal matrices, which is important so that there is no bias in the generation of the matrices that an attacker can use. The Haar distribution is a uniform measure over orthogonal matrices, with the property that if $Q$ is an orthogonal matrix sampled from the Haar distribution and $U$ and $V$ are any orthogonal matrices, then $UQV$ is also a sample from the Haar distribution [38]. Note that this property of orthogonal matrices from the Haar distribution implies the following useful property: given a transformed weight $W' = TW$, then all candidates for $T$ are equally likely, so there is no way to distinguish between this decomposition of $W'$ and any decomposition of the form $W' = (TV)(V^TW)$ where $V$ is any orthogonal matrix. Thus, while the singular values and the right singular vectors are unchanged, the left singular vectors are sufficiently scrambled giving no useful knowledge of $W$.

A standard way to generate a matrix from this distribution is to use the QR decomposition on matrices with elements from the standard normal distribution [38]

$$Q,\ R = \texttt{torch.linalg.qr(torch.randn(d,d))} \tag{65}$$
$$T = \texttt{Q * torch.diag(R).sign()} \tag{66}$$

where we use PyTorch [40] code.

We also experimentally confirm that these transforms scramble with high probability. We generate orthogonal matrices $T_i$ from the Haar distribution, and multiply them with random normal generated weights: $W' = WT_1...T_n$ where we vary $n$. We find that the distribution of the $W'$ are such that its relative Frobenius norm

$$\frac{\|W - W'\|_F}{\|W\|_F} \tag{67}$$

is tightly distributed around $\sqrt{2}$, and the cosine similarity of the flattened weights (where flat$(\cdot)$ is the flattening operation)

$$\frac{\langle \text{flat}(W), \text{flat}(W') \rangle}{\|\text{flat}(W)\|_2 \|\text{flat}(W')\|_2} \tag{68}$$

is tightly distributed around 0. This indicates that in practice the transforms are unlikely to align with a given weight matrix, instead transforming it to an almost orthogonal direction.

**Low condition number matrices** In order to generate matrices with higher frequencies yet small condition number, we generate matrices of the form $UDV^T$ where $U, V$ are Haar orthogonal matrices and $D = \text{diag}(d)$ is such that $d_i = se^{u_i}$, $u_i \sim \mathcal{U}[-\varepsilon, \varepsilon]$. Thus the singular values, which are the entries of $d$ as this is in SVD form, are within $[se^{-\varepsilon}, se^{\varepsilon}]$ and the condition number is bounded by $e^{2\varepsilon} (\approx 1 + 2\varepsilon)$. As the determinant, which is $\approx s^d$, scales the weights, we force the weight scale to cycle by cycling $s$.

# F Optimization

Let us denote transformation of components of the network from its original space to the one determined by a transform $T$ applied to weights by the subscript $_{(T)}$. Then we note that the transformation of the weights $W$ and the gradients of the weights $G$ are not the same:

- **Weight Space Transformation**: $W_{(T)} = WT$
- **Gradient Space Transformation**: $G_{(T)} = GT^{-T}$
  To see this note $G = \nabla_W \mathcal{L}$, $G_{(T)} = \nabla_{W_{(T)}} \mathcal{L}$, so

$$G = \nabla_W \mathcal{L} = \left(\nabla_{W_{(T)}} \mathcal{L}\right)\left(\nabla_W WT\right) = G_{(T)} T^T \tag{69}$$

which implies $G_{(T)} = GT^{-T}$.

Note that **we do not need to compute the transformed gradient $G_{(T)}$ explicitly**, if we use the transformed weight $W_{(T)}$ in the forward pass, in the backward pass automatic differentiation will automatically calculate $G_{(T)}$.

## F.1 Stochastic gradient descent variants

Most variants of stochastic gradient descent (SGD) use an update of the form

$$W^{(t+1)} = W^{(t)} - \eta f(G^{(t)}, G^{\text{old}}) \tag{70}$$

where $G^{(t)}$ is the current gradient and $G^{\text{old}}$ is saved information from previous gradients (e.g. $n^{\text{th}}$ moments of the past gradients). Thus there are two alternatives we can have which may not necessarily be the same, either we can ensure the parameter updates are equivalent to parameter updates in the original space

$$W^{(t+1)} = \left(W^{(t)} - \eta f(G^{(t)}, G^{\text{old}})\right)_{(T)} \tag{71}$$

$$= \left(W^{(t)} - \eta f(G^{(t)}, G^{\text{old}})\right) T \tag{72}$$

or we can ensure that every part of the computation uses the transformed result

$$W^{(t+1)} = \left(W^{(t)}\right)_{(T)} - \eta f\left(\left(G^{(t)}\right)_{(T)}, G^{\text{old}}_{(T)}\right) \tag{73}$$

$$= W^{(t)} T - \eta f\left(G^{(t)} T^{-T}, G^{\text{old}}_{(T)}\right). \tag{74}$$

Furthermore, it might not be trivial to compute $G^{\text{old}}_{(T)}$ from $G^{\text{old}}$. The most common saved gradient information is an estimate of the first moment of the gradients, $m = \mathbf{E}[g]$, for which $m_{(T)} = mT^{-T}$. However Adam/AdamW also use the second moment $v = \mathbf{E}[g^2]$, for which $v_{(T)}$ cannot be computed from just $v$. The issue here is that sign information is lost, so we cannot apply the transformation. As a result, we stick to optimizers that only use $m$, e.g. SGD + Momentum.

In our implementation we use Muon, which also only uses $m$. Furthermore we use Equation (73), which is easier to implement, rather than Equation (71). However, for Muon (and many other optimizers), if we use orthogonal transforms $T$ then Equation (71) and Equation (73) are equivalent, with the key point being that $T^{-T} = T$ for orthogonal matrices.

## F.2 Pure SGD

In pure SGD, we take steps of the form

$$W^{(t+1)} = W^{(t)} - \eta G^{(t)}. \tag{75}$$

**If we transform our weights before the forward pass:** our weights become $(W^{(t)})_{(T)} = W^{(t)} T$, and our gradient becomes $G_{(T)} = GT^{-T}$, so our step is

$$W^{(t+1)} = (W^{(t)})_{(T)} - \eta (G^{(t)})_{(T)} \tag{76}$$

$$= W^{(t)} T - \eta G^{(t)} T^{-T}. \tag{77}$$

Note this is a valid step, we are just applying SGD to a different network.

**What would the step be if we did the step in original space and then transformed to the new weight space:**

$$W^{(t+1)} = \left(W^{(t)} - \eta G^{(t)}\right)_{(T)} \tag{78}$$

$$= \left(W^{(t)} - \eta G^{(t)}\right) T \tag{79}$$

$$= W^{(t)}T - \eta G^{(t)}T. \tag{80}$$

**What correction term would make it equivalent**: we would need to transform our gradients by $T^T T$

$$W^{(t+1)} = (W^{(t)})_{(T)} - \eta (G^{(t)})_{(T)} T^T T \tag{81}$$

$$= W^{(t)}T - \eta G^{(t)} T^{-T} T^T T \tag{82}$$

$$= W^{(t)}T - \eta G^{(t)}T. \tag{83}$$

**Issue with this correction:** Note however this is a one transform correction. If we are just now applying the $n^{\text{th}}$ transform, normal gradient descent would be

$$W^{(t+1)} = W^{(t)}T(1)...T(n) - \eta G^{(t)}T^{-T}(1)..T^{-T}(n) \tag{84}$$

and to make the step equivalent to doing the step in the original space we would need to store $Q(n) = T(1)..T(n)$ and then apply

$$W^{(t+1)} = W^{(t)}T(1)...T(n) - \eta (G^{(t)}T^{-T}(1)...T^{-T}(n))Q^T(n)Q(n). \tag{85}$$

However storing $Q(n)$ means we have an exact way to undo the transforms on our weights: $W_{(T)} = WT(1)...T(n)Q^{-1}(n)$.

### F.3   SGD + Momentum

In SGD + momentum, we take steps of the form

$$W^{(t+1)} = W^{(t)} - \eta \left(\alpha m^{(t-1)} + \beta G^{(t)}\right). \tag{86}$$

**If we transform our weights before the forward pass:** our weights become $(W^{(t)})_{(T)} = W^{(t)}T$, and our gradient becomes $G_{(T)} = GT^{-T}$, so our step is

$$W^{(t+1)} = (W^{(t)})_{(T)} - \eta \left(\alpha m^{(t-1)} + \beta (G^{(t)})_{(T)}\right) \tag{87}$$

$$= W^{(t)}T - \eta \left(\alpha m^{(t-1)} + \beta G^{(t)}T^{-T}\right) \tag{88}$$

which is not a valid step as the previous momentum term is not in the right space.

**Fixing the momentum to be in the right space:** we need to apply the gradient space transform $T^{-T}$ to the momentum too

$$W^{(t+1)} = (W^{(t)})_{(T)} - \eta \left(\alpha m^{(t-1)}_{(T)} + \beta (G^{(t)})_{(T)}\right) \tag{89}$$

$$= W^{(t)}T - \eta \left(\alpha m^{(t-1)}T^{-T} + \beta G^{(t)}T^{-T}\right). \tag{90}$$

**Correcting this to be equivalent:** we would need to apply $T$ to the momentum (not doing the above correction) and $T^T$ to the gradient:

$$W^{(t+1)} = (W^{(t)})_{(T)} - \eta \left(\alpha m^{(t-1)}T + \beta (G^{(t)})_{(T)}T^T\right) \tag{91}$$

$$= W^{(t)}T - \eta \left(\alpha m^{(t-1)}T + \beta G^{(t)}T^{-T}T^T\right). \tag{92}$$

Note however that no correction steps are needed if $T$ is orthogonal, as then $T = T^{-T}$.

## F.4 Muon

In Muon, we take steps of the form

$$W^{(t+1)} = W^{(t)} - \eta O \left( \alpha m^{(t-1)} + \beta G^{(t)} \right). \tag{93}$$

where $O(M)$ is an efficient approximate orthogonalization procedure [19]. The non-approximate orthogonalization procedure is the mapping $USV^T \mapsto UV^T$.

**If we transform our weights before the forward pass, and apply our momentum correction term before the optimization step:**

$$W^{(t+1)} = (W^{(t)})_{(T)} - \eta O \left( \alpha m^{(t-1)} T^{-T} + \beta (G^{(t)})_{(T)} \right) \tag{94}$$

$$= W^{(t)}T - \eta O \left( \alpha m^{(t-1)} T^{-T} + \beta G^{(t)} T^{-T} \right). \tag{95}$$

**If $T$ is orthogonal:** Then $T^{-T} = T$, and since $\text{SVD}(M) = USV^T$ implies that $\text{SVD}(MT) = US(V^T T)$, then this is equivalent to original updates in the space.

# G  Matrix System attack

Applying transforms to the weights means that the intermediate activations $X_i(t)$ have been transformed. Thus one way to determine a bridge matrix is to use the intermediate activations.

Let us assume that the attacker has access to stage $i$ and $i + 1$ at time $t$ and $t' > t$, so they have $V_i(t)$, $T_i(t)$, $X_i(t)$ from stage $i$ and $T_i(t')$, $X_i(t')$, $U_{i+1}(t')$ from stage $i + 1$. Note that

$$X_i(t') = X_i(0)T_i(1)...T_i(t)T_i(t+1)...T_i(t') \tag{96}$$

so we have that

$$X_i(t') = X_i(t)\hat{T} \tag{97}$$

where $\hat{T} = T_i(t+1)...T_i(t')$ is the bridge matrix between times $t$ and $t'$ for the boundary between stage $i$ and stage $i + 1$. This means that $\hat{T}$ is the solution of a matrix system.

However in reality the $X_i(t)$ change over time steps $t$ for reasons other than the transforms $T_i(t)$. This is due to (1) different initial inputs $X_0(t)$ being used at different time $t$, and (2) when training, over different time steps the weights change.

We now investigate these two issues: the requirements the attacker needs to have this matrix system be defined, and whether the attacker can determine the bridge matrix from the matrix system.

**Requirements for the matrix system.**  The assumption that the same initial data was inputted to the network at both time steps is impractical: we are serving inference or training from a very large dataset where data is randomly sent to different pipelines, so the chances of the same initial data being inputted to the network are extremely small.

However the important requirement in the attack was that the intermediate activations to stage $i$ were the same at both time steps, which an attacker can force. In order to do this, the attacker requires an additional stage $j \leq i$ at both $t$ and $t'$. Then at both time steps, the attacker can output equivalent false activations: at time step $t$ they can output $FV_j(t)$ and at time step $t'$ they can output $FV_j(t')$ where $F$ is a fixed full rank matrix. As long as $X_i(t)$ ends up being full column rank, then this can work.

However this is also easy to defend against: we can check if the activations outputted from a node follow a similar distribution to activations at previous time steps, and also check against the distribution of activations of the same stage in other replicas. The attacker is likely to be caught and expelled between time step $t$ and $t'$ (as we have put in place measures to ensure $t' >> t$).

**Determining $\hat{T}$ from the matrix system.**  We consider the matrix system in Equation (97). Note that $X_i(t)$ and $X_i(t')$ are of the same shape $b \times d$ where $b$ is the batch size and $d$ is the embedding dimension, while $\hat{T}$ is $d \times d$.

If $X_i(t)$ has full column rank then $\hat{T}$ can be uniquely determined by $(X_i(t))^+ X_i(t')$ where $^+$ indicates the Moore-Penrose pseudo-inverse.

If $X_i(t)$ does not have full column rank, then there are infinitely many solutions. To bias towards a specific solution we can use least squares, or if we know that $T$ is orthogonal we can use orthogonal Procrustes. This is likely to only give a good solution if $X_i(t)$ is close to full column rank.

## H  Further Experimental Details

We use PyTorch [40] for our implementations. Since our framework is simple, we implement it within three frameworks, torchtune [58], torchtitan [34] and NanoGPT [24]. We use torchtune for our inference time experiments, torchtitan for our learning attack experiments, and NanoGPT (and its derivatives, discussed below) for the training experiments.

When generating and folding in our transforms, we do this in float64, which is one of the reasons that the morphing step is so expensive. This turns out to be very important for generating orthogonal matrices (doing the QR decomposition at a lower precision causes the error in $Q^T Q - I$ to be large). Furthermore to reduce floating point error, when folding we first cast the weights to FP64, multiply in our transforms, and then cast back down to the original precision of the weights.

For the learning attack experiments, we use the default hyperparameters in torchtitan. We grid search on the learning rate between torchtune's learning rate for finetuning Llama, 2e-5, to torchtitan's learning rate for from scratch training Llama, 3e-4. We find that for a few inconsistent boundaries, the optimal learning rate is closer to finetuning, while for many inconsistent boundaries (which we show results on) the optimal learning rate is closer to from scratch training.

For the training experiments we use the Muon speed-runs of NanoGPT [18]. However since the architecture has diverged away from the standard transformer architecture over time, we use a re-implementation of a specific speed-run milestone where the architecture is the closest to modern transformers [48].

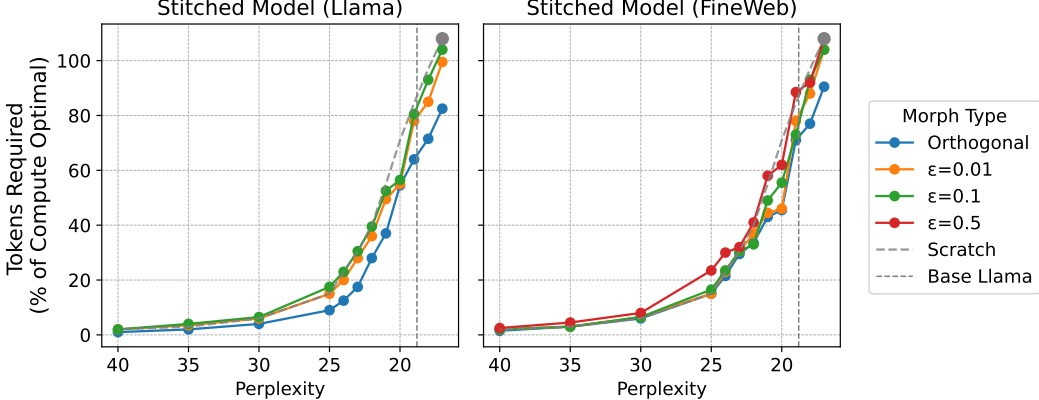

Figure 6: Tokens required, as a percentage of compute optimal (20B tokens), to reach various perplexities on FineWeb with a stitched Llama 3.2 1B model. We evaluate this with two sets of pretrained model weights that are morphed and then stitched: the base Llama model weights released by Meta (which is trained on a wide variety of data) and our model weights that we train from scratch on FineWeb. Transforms are applied such that the boundary between transformer layers are inconsistent.

## I  Further Results

In Section 5.4 models were trained to reach a perplexity of 20, which is similar to the perplexity of the original base Llama model released by Meta (roughly 19). Here we continue running these models (learning based attacks and their baselines) to a perplexity of 17, which is enough for the baseline to reach compute optimal (20B tokens for our 1B parameter Llama model), updating Figure 5 to Figure 6. We represent the tokens required as a percentage of compute optimal.

Here, training with orthogonal required at least 80% of compute optimal (roughly 80% of baseline compute) to reach a perplexity of 17, and $\varepsilon > 0$ transforms required roughly the same compute as the baseline.

## J   Feasibility of Decentralized Training

We briefly review the literature in communication efficient training, noting that several works have made progress towards large-scale model training in the decentralized setting and that progress here may be more advanced than many researchers are aware.

Communication efficient DDP is effectively solved, as it is a side effect of much of the federated learning literature. Gossip protocols can be used in place of the synchronous all reduce [7, 5], or simply many inner steps can be taken for each outer step [11]. Convergence guarantees can be obtained in this setting [31, 33] even with the communication graph altering during training [55, 26].

Moshpit-SGD [50] introduces a notable approach that is both communication efficient and scales well with heterogeneous compute and communication bandwidth. Diskin et al. [10] perform a real run, over 200 MB/s interconnects, using devices with a range of capabilities on a dynamic swarm to train a 72.5M parameter ALBERT-xlarge variant. Here a community of volunteers completed a training run that would have taken over 3 months using a standard 8xV100 setup, within 22 days on consumer hardware. Nodes were verified with an allow-listing approach, all gradient computations are assumed to be correct and each node computes full model gradients and hence has a copy of model weights.

Large models are also possible, with SWARM parallelism [51] showing pipeline parallel training becomes *less* communication intensive relative to compute as models grow larger; hence improving utilization. This work practically demonstrated training of a 1B parameter LLM on T4 GPU's with 500 MB/s interconnects, achieving roughly 20% throughput overhead to centralized training, maintaining very high accelerator utilization, and also possessing basic fault tolerance. This is achieved with redundancy within each pipeline stage, and dynamic and elastic routing between each stage, and the assumption of good actors. Learning@Home [49] propose the Decentralized MoE architecture and an asynchronous training scheme in order to achieve communication efficiency over heterogeneous nodes. Such an approach can theoretically scale to very large parameter sizes but has not been practically demonstrated beyond 257M, and while node failures are handled, byzantine nodes are not.

Recently, Avraham et al. [4] performed a public, large-scale, pipeline-parallel, decentralized training run at the 7.5-billion-parameter scale. This run was based on the SWARM [51] framework and used the technique from Ramasinghe et al. [47] to losslessly compress activations and their gradients.

