# OpenReview forum: "Unextractable Protocol Models: Collaborative Training and Inference without Weight Materialization"
_NeurIPS.cc/2025/Conference — NeurIPS 2025 poster_

### Official Review · Reviewer_Ed6f · 2025-06-25

**Clarity:** 2
**Significance:** 3
**Originality:** 4
**Rating:** 4
**Confidence:** 4

**Summary:**

This paper introduces Unextractable Protocol Models, a framework for collaborative training and inference of large neural networks where participants each hold only a subset of model parameters, preventing any single entity from extracting the complete model weights.
The approach periodically applies time-varying, random, invertible transforms at participant boundaries that preserve network functionality but render cross-time parameter assemblies incoherent and mitigate attacks that attempt to collect model shards across different time steps. Experiments demonstrate that the method maintains model performance while adding minimal computational overhead and significantly increases the cost for attackers to reconstruct full models.

**Questions:**

- Why is it that you need double precision to compute the matrices $Q$? Numercial stability? If so, where does this come from? Can you please elaborate on these dynamics?

**Ethical Concerns:**

["NO or VERY MINOR ethics concerns only"]

**Final Justification:**

The authors have addressed my concerns.

**Limitations:**

Sufficiently discussed.

**Paper Formatting Concerns:**

None.

**Quality:**

3

**Strengths And Weaknesses:**

**Strengths**:
- The paper addresses an interesting problem where, instead of keeping data private, the model has to be kept private and inaccessible by participants. This could be relevant in cases where a model learns patterns and behavior that must not be made public.
- The contributions are well described and nicely distinguished from related work, espeically federated learning


**Weaknesses**
- While I appreciate the approach and think that model security is an important topic, the abstract appears overly specific to me. I think, the abstract would benefit more from a discussion _why_ model secrecy is key in future decentralized applications.
- Similarly, the introduction does not discuss _why_ model secrecy is practically relevant. I was also wondering what the incentive for a participant would be to train a model they do not benefit from? In Federated Learning, clients at least get to benefit from the global model and their peers' data.
- To me it is not fully clear how the level of model secrecy interacts with the number of transforms, i.e., what level is considered as "secret" or how do you compute likelihood for a successful model extraction? This question is most fundamental in my review.
- When doing a speed run comparison, showing the loss or perplexity trajectory over time is usually beneficial for the reader to understand whether the proposed approach is viable for their individual use cases.

---

> ### Author Rebuttal · Authors · 2025-07-29
>
> We thank the reviewer for their acknowledgement of the clarity of the setting and the differentiation to related work
>
> ### **W1,W2: Understanding Model Secrecy**
> Model secrecy is key in decentralized models as it is **the only way to make the economics of large scale decentralised training practical**. The fundamental problem we address is that today, there is no economic incentive for participants to collectively donate the millions of dollars worth of compute required to train a very large and expensive model if one of them can easily leak the trained weights (explained on lines 22-27). The solution is for a scheme that allows for such a model to be collaboratively trained and hosted while no participant can gain access to the entire weight set (lines 28-29). This allows schemes for programmatic value flow (for example, assigning revenue to participants) to be implemented. We note that reviewer kk7D praises this setting as "extremely realistic and useful". We will make the setting, its importance, and differences to existing settings such as federated learning more pronounced in the improved version.
>
> ### **W3: Model Secrecy and Number of Transforms**
> If a simple static model is hosted, which anyone can join and contribute to hosting, an attacker can gain access to the network and collect weights from different stages over time, eventually assembling the full model. At a high level, our approach prevents this by making the model weights incompatible with each other at different time steps. Thus **model security is not tied to the number of transforms**, but to ensuring that an attacker cannot gain a set of weights that use the same transforms, which requires **ensuring that a new transform is applied between when an attacker exits the protocol and enters again**. Note that this can be ensured by making participants have to wait for a transform step before entering, so there is a trade-off between the join rate and the overhead of the transforms (as more frequent transforms require more computation). As mentioned on line 270, our practical implementation has a transform step every 30s, meaning that anyone joining the protocol has to wait at least 30s, and has an overhead of 3% to total latency.
> We provide a much more detailed write up of the setup and the probability of extraction for the attacker (both with and without UPMs) in Appendix A.
>
> ### **W4: Loss Curves**
> As mentioned in lines 323-326, **for orthogonal transforms the loss curve is exactly the same as the original loss curve**. We demonstrate this in Figure 6 in the appendix, where we plot the difference between the original loss curve and our approach over time, showing that for orthogonal transforms with on GPT2 124M, 355M, and 1.5B, the difference between the loss curves at each step is roughly 1e-2 (the final loss is 3.28) over 1.75B tokens. This highlights the importance of our decision to use orthogonal transforms during training; orthogonal transforms make the transformation on gradient space also orthogonal, so the exact same steps are taken (which is mentioned in Section 3.3, proven in Appendix D, and experimentally shown in Figure 6).  To highlight the extent to which our approach does not alter training, we include the loss curve as a table below (data from the same source used to plot Figure 6.).
>
> | Method         |   1    |  500   |  1000  |  1500  |  2000  |  2500  |  3000  |  3500  |  3611  |
> |---------------|--------|--------|--------|--------|--------|--------|--------|--------|--------|
> | NoTransforms   | 11.013 | 3.986  | 3.528  | 3.408  | 3.407  | 3.444  | 3.241  | 3.145  | 3.273  |
> | Our Approach     | 10.979 | 3.993  | 3.524  | 3.420  | 3.408  | 3.445  | 3.241  | 3.139  | 3.268  |
>
> ### **Q1: Double Precision for Computing Transforms**
> Double precision is needed to properly generate and represent the transforms, as otherwise there is large floating point representation error. Generating random orthogonal matrices as per Eq. 52-53 in Appendix C2 with single precision (compared to double precision) drastically reduces the orthogonality of the resulting transforms (how large $||Q^TQ-I||_2$ is). After generating transforms in double precision, the transforms can be combined with single precision weights and the output stored in single precision (that is, the precision of the actual weight matrices is never altered), and this is sufficient to avoid numerical precision errors as shown in Figure 3.

---

### Official Review · Reviewer_Lkyy · 2025-06-29

**Clarity:** 3
**Significance:** 3
**Originality:** 3
**Rating:** 5
**Confidence:** 3

**Summary:**

The paper proposes addresses the issue of model stealing in decentralized collaborative training setup where each party contributes training of a part of a large model. An attacker may join such collaborative training at different times and accrue the entire model parameters. The paper proposes to morph invertible transforms in consecutive model stages such that in a given time-step these transforms cancel each other (keeping model outputs the same as without transforms) but in different time steps they do not (changing the model outputs). This way an adversary cannot just join the collaborative training at different times to steal the model. They evaluate their protocol on models with ~1B params, discuss the computation and latency overhead, and potential attacks against their protocol.

**Questions:**

- Do you need to apply transforms for each stage for this defense to work? Can you omit a few complex stages and still get the same protection from model stealing?
- Is it necessary to apply complex matrix based transforms or simple linear transforms can suffice? For instance, what if each stage add a random time-dependent noise to its output and subtract the same from input of the next stage? This might also reduce the complexity in backward pass as all noise is just a constant.

**Ethical Concerns:**

["NO or VERY MINOR ethics concerns only"]

**Final Justification:**

The rebuttal address my concerns and I would like to maintain my rating of Accept for this paper.

**Limitations:**

- Perplexity increase is not trivial for fp16/bf16, which can be concerning.

**Paper Formatting Concerns:**

No concerns.

**Quality:**

3

**Strengths And Weaknesses:**

Strengths
- Idea of morphing time-dependent transforms is very simple and seems effective in most cases. Given the growing sizes of modern ML models, this can be a simple and effective method to thwart model stealing.
- The paper does a nice job of explaining the idea. It is well-written and easy to understand.
- Evaluations show that morphing does not impact functionality of the model in most cases and the latency overhead at inference time is minimal.
- The paper provides a comprehensive set of subfunctions to be used for various common functions involved in modern LLMs.

Weaknesses
- The motivation of the attacker is not clear to me. If the attacker can host an entire model and  that the final trained model will anyways be available for use by everyone, can the attacker not just steal the final model? Why does the attacker want the intermediate models?
- Perplexity increase is not trivial for fp16/bf16, which can be concerning.

Minor:
- Line 130: should be figure 2A.

---

> ### Author Rebuttal · Authors · 2025-07-29
>
> We greatly appreciate the reviewer's characterization of our method as "simple and effective," and their remark that the manuscript is "well-written and easy to understand".
>
> ### **W1: Attacker Motivation**
> The final trained model is not intended to be accessible by everyone, **even at inference**, only participants that hold part of the model can see their portion.
> As a result, after training the model exists, and yields value, only if enough participants continue hosting model portions to support inference.
> The model's value would be lost if the model was released after training (like current open weight models), or leaked by attackers. Note that **even partially trained versions of the model can still have decent performance**, and thus value, so it is important to also protect against weight extraction during training.
>
> ### **W2: Perplexity Increase**
> These experiments are provided to point out **an important consideration when implementing UPMs**. During training, gradient descent naturally corrects any floating-point drift introduced by numerical imprecision from the morphing, as demonstrated by the results in Table 1 and Figure 6. For low-precision inference, a workaround would be to actually morph high-precision weights (FP32 or FP64) stored on disk, which are then cast down for a low-precision copy in GPU VRAM used for inference. Note that our results in Figure 3 show that perplexity increase is extremely slow for FP32. Because transforms are applied infrequently, disk-to-VRAM communication is negligible, and the on-device memory holds only the low-precision weights.
>
> ### **W3: Typos**
> Thanks, we will fix this in the improved version.
>
> ### **Q1: Transforming All Stages vs Some Stages**
> Yes, it can work for only transforming a few stages, however as the weights are now less "scrambled", this becomes a trade-off with how much compute an attacker needs to pull off a learning attack. It is quite likely that a better trade-off can be gained by transforming more complex stages as you suggest. We leave investigating this trade-off to future work.
>
> ### **Q2: Possibility of Simpler Transforms**
> Adding random time-dependent noise to a layer's output does not change the weights, it only changes the activations. Thus weights are not inconsistent over time and can easily be pieced together to get the full model.
> We specifically targeted complex and random matrix based transforms over simple linear transforms in order to make learning based attacks difficult. It is conceivable that transformations such as simple scaling of the entire weight matrix would be easy to undo by gradient descent.

---

> > ### Comment · Reviewer_Lkyy · 2025-08-05
> > **Thanks for the response!**
> >
> > Thanks for the responses! They address my concerns and I would like to maintain my score and believe that paper should be accepted.

---

### Official Review · Reviewer_vesi · 2025-07-02

**Clarity:** 3
**Significance:** 2
**Originality:** 3
**Rating:** 4
**Confidence:** 3

**Summary:**

This paper proposes Un-extractable Protocol Models (UPM), a model parallel based training setup under decentralized setting to provide a solution for full model attack under the distributed setting. And authors provide experiment evaluation on GPU devices to verify the proposed method on Llama 3.2 1B on Functional Equivalence, inference overhead, and training quality.

Contribution:
- introduce Unextractable Protocol Models (UPMs), a framework enabling collaborative training and inference without allowing any participant to extract the full model weights.
- Use experiment to verify the inference overhead, and training quality.

**Questions:**

- The setting, is it a new one?
- Why the setting is a reasonable setting?
- Especially why this kind of attacker is reasonable here?
- How about other solutions for the current setting? Why the proposed one?
- how to synthesize the distributed setting here, and why this kind of experiments can really evaluate the latency in real distributed setting
- Since evaluated on GPU environment, does the setting of hyper parameters (for example, batch size) would impact the overall latency? Does these conclusion changes when batch size changes?

**Ethical Concerns:**

["NO or VERY MINOR ethics concerns only"]

**Final Justification:**

The authors have addressed most of my concerns and I will maintain my support to the paper.

**Limitations:**

Yes

**Quality:**

3

**Strengths And Weaknesses:**

- __Strengths__
  - Clarity: the paper is cleared written, especially providing throughout description on the proposed model architecture
  - Quality: meaningful solution and design.
  - Originality: Good. the paper gets insights from model parallel to propose such framework for distributed setting.

- __Weakness__
  - Limited discussion on why previous work is not sufficient for targeted setting
  - Unclear about the real-world applicable setting for mentioned attacker. Why it is a readable one?
  - Limited description for detailed experiment setting, say, how to synthesize the distributed setting here, and why this kind of experiments can really evaluate the latency in real distributed setting

---

> ### Author Rebuttal · Authors · 2025-07-29
>
> We are grateful to the reviewer for praising the paper as "clearly written" and recognizing it is a meaningful solution.
>
> ### **W1,W2,Q1-Q4: Understanding the setting, previous work, and other solutions**
> We wish to clarify that this is a new setting (see line 32), which reviewer kk7D praised as "extremely realistic and useful".
> Decentralized training is an early but rapidly growing sub-field (see Appendix H for a full discussion of emerging work).
> The problem our work addresses is that the real world economics of large scale decentralised training are not practical, as there is no incentive for participants to collectively donate the millions of dollars worth of compute required to train a very large and expensive model **if one of them can easily leak the trained weights** (explained on lines 22-27). The solution is for **a scheme that allows for such a model to be collaboratively trained and hosted while no participant can gain access to the entire weight set** (lines 28-29). In short, unmaterializability lets models be created, hosted, and used solely within the protocol, enabling incentive mechanisms such as revenue-sharing among participants. If any participant can obtain the full model, these incentives become impossible. We will make the setting and its novelty more clear in the improved version.
>
> As this is a novel setting there is no directly comparable prior work. As discussed in Section 4, the closest related work is black-box extraction attacks, which aim to gain a model with similar capabilities using only API queries. Other potential solutions for our setting, discussed on line 97, either rely on centralization or on expensive cryptographic primitives such as secret sharing or homomorphic encryption. Our method exploits the inherent structure of distributed pipeline parallel training, making it much more efficient. We also discuss the differences and similarities between our setting, Federated Learning and Decentralized Federated Learning in the related work (Section 4).
>
> ### **W3,Q5,Q6: Evaluating latency**
> There has been several recent works on feasibility of decentralized training [8,36,37] (see Appendix H for a full discussion) which focus on latency and throughput numbers that can be achieved in this setting.
> Our evaluation in Section 5.2 shows that the overhead of UPMs in this setting is quite low during inference (e.g., 3% impact to overall latency and 0.1% impact to total communication volume), the phase where morphing's impact is most pronounced. During training, the relative impact on all metrics decreases because memory use, communication, and computation grow with the model’s operations.
>
> These figures depend on both **the morphing frequency and the communication efficiency of the distributed implementation**. Although the frequency can be reduced, the system must prevent participants from rejoining within a morphing interval. We did our communication overhead calculations relative to our centralized implementation, so the overhead of our method in practice could be much lower. Other hyper-parameters may influence communication efficiency, but to a much lesser extent.

---

> > ### Comment · Reviewer_vesi · 2025-08-05
> >
> > I would like to thank the authors for responding to my comments and I will maintain my support to the submission.

---

### Official Review · Reviewer_kk7D · 2025-07-02

**Clarity:** 3
**Significance:** 3
**Originality:** 4
**Rating:** 5
**Confidence:** 3

**Summary:**

The authors propose a new collaborative training framework, entitled Unextractable Protocol Models (UPMs). The framework ensures that adversarial participants cannot extract the full model weights by joining the protocol in consecutive stages, under different aliases, in order to stitch portions of the model together to recover the entire model. Analysis and experimental validation of UPMs showcase that they retain accurate outputs, even in the face of precision errors, while introducing time-varying random invertible transforms into the model weights at stage boundaries. This protocol will allow safer and more secure collaborative training and makes strides within the trustless,  decentralized setting.

**Questions:**

1. I was a little confused by Figure 4. How is it possible that the transformations actually reduce the number of tokens required to reach a perplexity of 20. There's no free lunch, especially when the transformations are in some sense a barrier to true performance.
2. How many morph steps in total are used during training? If each iteration is one morphing step (from what I've gathered from Line 139), aren't tens of thousands of iterations common in training an LLM from scratch?
3. Relating to Q2, if the number of morph steps is quite large, there does seem to be a sizable decrease in model performance for certain LLMs. First, how much degradation is there for a normal training or fine-tuning session? Second, would UPMs no longer be viable for large-scale training at smaller precision? Thus, UPMs likely cannot be utilized in low-memory settings.
4. Is there still any valuable information that an adversarial agent can glean from picking up different portions of the model, even if they have different transformations? For example, would size, magnitude, or distribution of the values throughout the weights be sensitive information?

**Ethical Concerns:**

["NO or VERY MINOR ethics concerns only"]

**Final Justification:**

Very original and interesting paper with great results. It’s an accept from me! Authors did a wonderful job in their rebuttals.

**Limitations:**

Yes.

**Quality:**

3

**Strengths And Weaknesses:**

# Strengths

1. Trustless, decentralized learning is an extremely realistic and useful area of research. The proposed protocol is an exciting idea that propels collaborative training a step forward.
2. The underlying idea, and proof-of-concept, is novel and well-worked.
3. The paper is clear, clean, and well-organized.
4. Experiments are detailed and well-done (outside of the weaknesses listed below).

# Weaknesses

1. Model performance does seem to degrade somewhat substantially when utilizing UPMs (Figure 3) at usual training precisions.
2. In the distributed setting, devices usually have low memory capacity and utilize low-precision training. As a result, UPM may be quite noisy when being utilized in resource-constrained settings (which are common to distributed settings).

---

> ### Author Rebuttal · Authors · 2025-07-29
>
> We deeply appreciate the reviewer's endorsement of our setting as "extremely realistic and useful", their description of the method as "an exciting idea that propels collaborative training a step forward", and their assessment of the experiments as "detailed and well-done".
>
> ### **W1: Performance degradation due to precision**
> UPMs **do not degrade model performance during training or inference in practice** (we will clarify this). We provide the experiment in Figure 3, which highlights the drift caused by finite precision, to **point out an important consideration when implementing UPMs**.
>
> During **training**, gradient descent naturally corrects any floating-point drift introduced by numerical imprecision from the morphing, as demonstrated by the results in Table 1 and Figure 6.
>
> During **inference**, we have two options.
> The first option is to run inference in high precision (FP32 or FP64) as Figure 3 shows that performance degrades extremely slowly for FP32, with perplexity increasing by less than $1\times 10^{-4}$ over 10,000 transform steps.
> The second option is to keep the final weights in high precision on disk while maintaining a low-precision copy in GPU VRAM for inference.
> Then, at each transform step we apply the high-precision transform to the high-precision weights on disk, which are cast down and loaded into GPU VRAM to replace the copy used for inference.
> Because transforms are applied infrequently, disk-to-VRAM communication is negligible, and the on-device memory holds only the low-precision weights.
> Note that our experiment in Figure 3 shows that such a workaround is important for low-precision inference as naive FP16/BF16 inference would slowly cause model weights to drift due to accumulating floating point error.
>
> ### **W2: Performance in resource-constrained settings**
> As discussed in Section 5.2, UPMs add minimal overhead for both communication and memory (given the inference workaround above). Only the transform must be communicated every few steps, which is smaller than an activation gradient, and once folded into a layer it can be discarded introducing minimal memory overhead.
> Furthermore, because UPMs operate in a pipeline-parallel setting where the model is split stage-wise across devices, even GPUs with limited VRAM (e.g., T4s) are already supported, as they need to hold only a single stage.
> For devices that are even more resource-constrained, stages can be further split and still be supported by our framework. For example, splitting a Transformer layer between the self-attention and MLP blocks would still work with our transforms, though we leave concrete experiments to future work.
>
> ### **Q1: Understanding Figure 4**
> The experiments in Figure 4 are **demonstrating learning-based attacks**, in which attackers have stitched together trained but incompatible weights (each with different transforms applied because they are from different time steps), rather than results from training models from scratch.
> Although incompatible weights cannot be combined to recover the original function, they still contain useful information.
> Consequently, a plausible attack is to fine-tune starting from these weights to determine whether a useful model can be obtained with less computation than training the entire model from scratch.
> Figure 4 shows that the computation required, measured as the number of tokens used during training, is less than training from scratch, with at most a 36% reduction.
> Extended results in Figure 5 in the appendix indicate that the reduction is at most 20% for performance matching compute optimal training.
>
> **For results on training from scratch** with and without UPMs, see Section 5.4 and Table 1.
> As expected, the number of tokens that UPMs require to match the perplexity of the baseline is the same or more (not less), and it specifically the same for orthogonal transforms.
> Additional plots are provided in Figure 6 in the appendices.
>
> ### **Q2: Morphing frequency during training**
> We apply transforms frequently during training (and inference) to prevent an attacker from leaving the protocol and rejoining within a single transform window.
> Thus, as mentioned on line 270, our standard is to morph every 30, which during training corresponds to a morph every 10-20 forward-backward iterations.
> If we want to ensure that no participant can exit and re-enter within those 30s, we can require each new participant to wait 30s before joining the protocol.
>
> ### **Q3: Performance degradation during training**
> As mentioned in the response to W1, during training gradient descent corrects for small precision errors leading to **no performance degradation**.
> We experimentally demonstrate this Table 1, training with orthogonal transforms yields no degradation in validation loss after a fixed number of iterations, whereas other transforms do as those transforms alter the loss landscape.
> We further show in Figure 6 in the appendix that the error in training loss (training drift) is small and constant throughout training with orthogonal transforms.
>
> ### **Q4: Information that can be gleaned by participants**
> Our method causes not only the weights but also the intermediate activations and gradients to change continually.
> Because we often use orthogonal transforms, which preserve weight and gradient norms, an adversary could still observe consistent statistics for those norms.
> However, there is little an adversary can do with that information.

---

> > ### Comment · Reviewer_kk7D · 2025-08-04
> >
> > Thank you for answering my questions. I maintain my score and believe the paper is worthy of acceptance.

---

### Official Review · Reviewer_mGZF · 2025-07-03

**Clarity:** 3
**Significance:** 4
**Originality:** 3
**Rating:** 5
**Confidence:** 4

**Summary:**

This paper address the important problem of collaborative machine learning by proposing a framework that enables both training and inference without allowing any single participant to access or extract the full model weights. From the reviewer's perspective, this approach to creating "unmaterializable" models is novel and addresses a significant challenge in collaborative machine learning.

**Questions:**

Regarding the experiments, have the authors noticed any effect on the experimental results when varying the number of attackers?

**Ethical Concerns:**

["NO or VERY MINOR ethics concerns only"]

**Final Justification:**

The authors' response addresses my primary concerns. I raise my score accordingly to reflect the paper's contribution in posing an important scientific question and its elegant method.

**Limitations:**

The authors do a good job of outlining the limitations of their work.
- They point out that while the morphing steps don’t mathematically change the forward function (or the inference time behavior), precision errors build up computationally when transforms are applied many times.
- They’ve analyzed the overheads for communication, inference, and memory.
- They also admit that their experiments are limited to single-machine simulations, leaving out real-world decentralized setups and side-channel attacks (like timing or cache-based ones).

**Paper Formatting Concerns:**

No formatting concerns identified.

**Quality:**

3

**Strengths And Weaknesses:**

**Strengths:**
- **Motivation:** The motivation is strong, diving into a critical, new, and underexplored area: unmaterializable weights, where no participant can obtain the full weight set.

- **Method:** The core idea of introducing an identity function between neighbors, and then decomposing it into a random transform and its inverse, is both simple and elegant.

- **Performance:** The experimental results are impressive. It is a strong practical result that the proposed protocol adds only minimal overhead—approximately 0.1% in bandwidth and 3% in amortized latency—while maintaining its security promises.

- **Transparency about Limitations:** The reviewer appreciate that the authors have a dedicated section clearly discussing the limitations of their work. They transparently acknowledge issues such as the fact that their experiments are limited to single-machine simulations.

**Weaknesses:**
- **Orthogonality under Nonlinear Self-Attention**: In Figure 3.1(c), the self-attention operation is nonlinear. How is the orthogonality of \( T_{i-1} \) preserved in this case? Since the transformation is not simply \( T_{i-1}T_{i-1}^{-1} \), but involves a nonlinear self-attention step, it is unclear how orthogonality is maintained. Could the authors provide a more detailed explanation of this point?
- **Clarity of Attack Descriptions:** The discussion of various attacks in Section 5.3 could be improved for clarity. To enhance readability, it would be helpful if the authors provided more background details on the specific attack methodologies they evaluated against. This could perhaps be included in an appendix to make the paper more self-contained for readers who may not be experts in this specific area.

---

> ### Author Rebuttal · Authors · 2025-07-29
>
> We sincerely thank the reviewer for praising our method and acknowledging that it "addresses a significant challenge in collaborative machine learning", characterizing this as a "critical, new, and underexplored area."
>
> ### **W1: Orthogonality under Nonlinear Self-Attention**
> We do not require the self-attention layer to be linear, we only need it to satisfy the definition of a valid subfunction (Equation 2) or its generalization (lines 150–175), in which _the function can be highly non-linear but the inputs interact linearly with the weights so that the transforms cancel out_.
> We explain explicitly on line 161 why self-attention qualifies as a valid sub-function.
> However, the normalization layer that precedes the self-attention function poses a challenge, as discussed on lines 176–191.
> A key contribution of our work is to show that a standard Transformer block with RMSNorm layers is a valid generalized subfunction under this definition when _orthogonal transforms are used for $T$_, and can therefore be morphed by our approach without changing the network's forward function.
> Concretely, we are able to pass the transform through the input norm, so $Norm(XT^{-1})TU=Norm(X)T^{-1}TU=Norm(X)U$, because its orthogonality ensures that the the normalization factor of RMSNorm does not change.
>
> ### **W2: Clarity of Attack Descriptions**
> Thank you for the suggestion.
> We already provide detailed discussions of each attack in the appendices: general **stitching attacks** with probability analysis in Appendix A, **matrix system attacks** with more realistic requirements in Appendix E, and **learning-based attacks** with more results in Appendix G.2.
> Following your suggestion, we will add a separate section in the appendix concisely summarizing these attacks.
>
> ### **Q1: Varying the number of attackers**
> For **stitching attacks**, Appendix A quantifies how varying the number of attackers affects the probability of successful extraction.
> For example, with 24 stages and 10 replicas, if attackers control 1% of the compute (e.g., 100 compromised nodes out of a total of 10,000 nodes) the probability of extraction within 10,000 steps is $2\times 10^{-7}$ %.
> For **matrix system attacks**, Appendix E details the requirements of a more realistic variant, which becomes easier to mount as the number of attackers increases (however the countermeasures are irrespective of the number of attackers).
> For **learning-based attacks**, the decisive factor is the attackers' aggregate compute, not their count. Figure 4 demonstrates experimentally how much compute they require relative to the cost of fully training the model (measured in training tokens).

---

> ### Comment · Reviewer_mGZF · 2025-08-06
> **Reviewer Response**
>
> Thank you for the response, which addresses my primary concerns. I will raise my score accordingly to reflect the paper's contribution in posing an important scientific question and its elegant method.

---

### Decision · Program_Chairs · 2025-09-17

**Decision:**

Accept (poster)

**Comment:**

This paper introduces Unextractable Protocol Models (UPMs), a framework for collaborative training and inference in which model shards remain incompatible across time, preventing any participant from reconstructing the full weight set. The method leverages pipeline parallelism and periodically applies invertible transforms at stage boundaries. Experiments on billion-parameter models show that UPMs preserve model accuracy, incur minimal training and inference overhead, and significantly raise the cost of model extraction attacks.

Reviewers found the paper novel, elegant, and clearly written. They highlighted the strength of the idea, the thorough analysis of attacks, the practical overhead results, and the relevance of the problem to decentralized machine learning. At the same time, they pointed out limitations: the approach applies only to pipeline parallel training, its guarantees are operational rather than cryptographic, the evaluation was carried out in single-machine simulations rather than real decentralized deployments, and inference drift occurs under low precision. The authors provided clarifications and mitigation strategies during the rebuttal, and reviewers were satisfied, with all ultimately recommending acceptance.

Beyond the reviewer discussion, I remain somewhat cautious about two aspects. First, the approach is essentially security by obfuscation, relying on repeated morphing to frustrate current extraction techniques rather than offering strong cryptographic guarantees. While this appears effective against the attacks considered, it may be fragile in the longer term. Second, the scope is narrower than the title suggests: the method is tied to pipeline-parallel setups and was evaluated only in simulated single-machine environments, not in a truly decentralized system. These caveats should be reflected more explicitly in the presentation.

Overall, despite these limitations, the work provides a timely and practical mechanism for collaborative training that has generated strong support from the reviewers.